# VIMO: A GENERATIVE VISUAL GUI WORLD MODEL FOR APP AGENTS

**Dezhao Luo[1*], Bohan Tang[2*], Kang Li[2], Georgios Papoudakis[3], Jifei Song[3], Shaogang Gong[1],**

**Jianye Hao[3], Jun Wang[4], Kun Shao[3†]**

[1]Queen Mary University of London, [2]University of Oxford,
[3]Huawei Noah's Ark Lab, [4]AI Centre, University College London

## ABSTRACT

App agents, which autonomously operate mobile Apps through GUIs, have gained significant interest in real-world applications. Yet, they often struggle with long-horizon planning, failing to find the optimal actions for complex tasks with longer steps. To address this, world models are used to predict the next GUI observation based on user actions, enabling more effective agent planning. However, existing world models primarily focus on generating only textual descriptions, lacking essential visual details. To fill this gap, we propose **ViMo**, the first **Vi**sual world **Mo**del designed to generate future App observations as images. For the challenge of generating text in image patches, where even minor pixel errors can distort readability, we decompose GUI generation into graphic and text content generation. We propose a novel data representation, the Symbolic Text Representation (**STR**), to overlay text content with symbolic placeholders while preserving graphics. With this design, ViMo employs a **STR Predictor** to predict future GUIs' graphics and a **GUI-text Predictor** for generating the corresponding text. Moreover, we deploy ViMo to enhance agent-focused tasks by predicting the outcome of actions. Experiments show that ViMo establishes visual world models as a compelling alternative to language-based approaches, producing visually plausible and functionally effective GUIs that empower App agents with more informed decisions. Project link: https://ai-agents-2030.github.io/ViMo/.

## 1 INTRODUCTION

Recent advancements in Large Language Models (LLMs) have unlocked new possibilities for deploying AI agents across diverse fields (Li et al., 2023; Gou et al., 2023; Rawles et al., 2024b; Zhou et al., 2025; Tang et al., 2025c;a; Zhang et al.; Ye et al., 2025; Xu et al., 2026). A notable application is the smartphone application (App) agents (Rawles et al., 2024a; Wang et al., 2024a; Beechey et al., 2025), designed to directly interact with Graphical User Interfaces (GUIs) to perform tasks autonomously and efficiently in a mobile operating system.

However, existing agents struggle with making decisions for tasks requiring longer steps (Chae et al., 2024). To address this "long-horizon" limitation, an increasing number of studies have introduced world models, which predict how GUIs evolve in response to user actions (Gu et al., 2024; Tang et al., 2025b). Yet, these models typically rely on language to describe future observations. These language-based descriptions often fail to capture the intricate visual details, such as the location and colour of GUI elements, necessary for a precise representation (Chae et al., 2024). A seemingly straightforward solution is to execute action candidates on App emulators. However, real-world execution is impractical for scalable planning since actions like payments or repeated purchases

---

\* Equal contribution.
† Corresponding author: shaokun2@huawei.com.
By LLMs, we refer to the concept of foundation models that accept various modalities (e.g., visual language models (VLMs), multimodal LLMs (MLLMs)) while producing textual sequences (W. contributors, 2024).

User Action:
Enter the Email as dbwscratch.test.id5@gmail.com

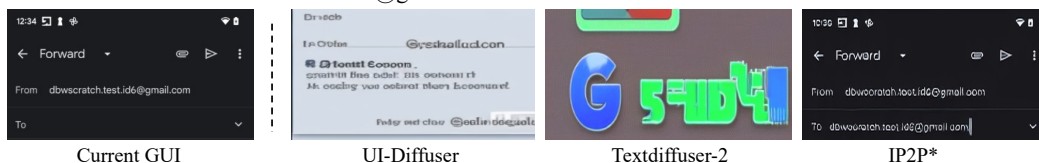

| Current GUI | UI-Diffuser | Textdiffuser-2 | IP2P* |

Figure 1: GUIs generated by image-based methods (UI-Diffuser (Wei et al., 2024), TextDiffuser-2 (Chen et al., 2024c), and IP2P (Brooks et al., 2023) fine-tuned on GUI dataset, denoted as IP2P*).

are difficult to backtrack. Similar concerns have motivated the broader world-model community to explore ML-based simulators (Li et al., 2025; Hafner et al., 2019; Hu et al., 2022). We tackle the problem in the GUI domain by designing a GUI world model capable of predicting hypothetical observations in the visual modality.

To build a visual GUI world model capable of generating plausible future GUI observations that are visually consistent with user actions, a straightforward approach involves generating each pixel of a GUI using image generations (Brooks et al., 2023; Rombach et al., 2022). Although these methods demonstrate promising results, such as the GUI graphic generation on the location, style, and colour of GUI elements (Wei et al., 2024), or scene-text generation in a style that aligns the visual context (Chen et al., 2024c; Zhang et al., 2024c), they still display distortions in the text rendering, particularly for small-sized text where each pixel is critical for accurately identifying and representing the text (see Fig. 1 for an illustration).

To address the challenges of accurately generating high-fidelity text content within a GUI, we propose **ViMo**, the first visual GUI world model. ViMo decouples the generation of graphic and text content into distinct processes, using a novel data representation named Symbolic Text Representation (**STR**). In STR, each text content is replaced (overlayed) with a text symbol, a rectangle-shaped placeholder with a defined border and fill colours, functioning as a special GUI element. Thus, we simplify the task of text content generation to text symbol generation, which reframes the problem to the localisation of the text within a GUI. Based on STR, ViMo employs a **STR Predictor** and a **GUI-text Predictor** to generate the graphic and the text content respectively. Specifically, the STR predictor is implemented as a diffusion model, taking the current STR, extracted from the given GUI, and a user action as inputs to generate the STR of the next GUI. Meanwhile, the GUI-text predictor, implemented based on an LLM, leverages the STR generated by the STR predictor to produce the corresponding text for each text symbol. Finally, the predicted STR and the generated text are combined to produce the next GUI.

We evaluated ViMo in three distinct scenarios to comprehensively demonstrate its effectiveness. First, we assessed its world model capability, where the quality of the generated GUIs was measured using visual similarity, instructional accuracy, and action readiness scores. Each score was examined through both automatic metrics and user studies. These assessments provided a robust and holistic evaluation of how visually precise and contextually plausible the generated GUIs were. Second, we tested ViMo in an agent-focused task to evaluate its benefits for existing App agents and its superiority over other language-based and image-based world models. In this setup, given a goal and the current App observation, the agent selected optimal actions to achieve the goal (Wang et al., 2024a). By accurately predicting the next GUI based on the current observation and an action, ViMo enabled the agent to better anticipate action outcomes and make more informed decisions. This experiment demonstrated the model's effectiveness in enhancing decision-making for App agents. Finally, we evaluated ViMo's real-world applicability under online navigation and zero-shot generalisation settings. These scenarios assessed the model's ability to perform in real-time interactions and to generalise to previously unseen Apps, further demonstrating its generalisation capabilities and practical value in dynamic environments.

Our main contributions are summarised as follows:

• We propose ViMo, the first generative visual GUI world model that predicts App observations in a visual modality, capable of more realistic and concrete visual GUI predictions compared to contemporary language-based methods.

• To address the challenge of strict pixel-level accuracy required to avoid distorted or blurred text generation in a GUI, we propose a Symbolic Text Representation (STR), overlaying text with uniform text symbols (placeholders) to simplify text content generation to text location generation. Then ViMo leverages an LLM to generate the corresponding text content for each text symbol.

• Extensive experiments demonstrated the effectiveness of ViMo in both world model evaluation and agent-focused tasks. Specifically, ViMo achieved an average 29.14% and 182.74% relative improvement over existing world models in terms of automatic metrics and user studies, respectively. Moreover, ViMo boosted the step-wise action prediction accuracy of App agents, achieving a 14.07% relative performance gain. In the online navigation setup, ViMo increased the task completion rate from 33.19% to 40.95%, yielding a substantial improvement of 7.76%.

## 2 RELATED WORKS

### 2.1 APP AGENT

App agents, powered by LLMs, have advanced task automation on mobile Apps (Wen et al., 2024b; Chen et al., 2024a; Zhang et al., 2024a;b; Lee et al., 2023). These agents interact with GUIs by emulating human actions. Approaches in this domain are broadly divided into *language-based* and *multi-modality-based* methods. Language-based methods rely on textual description of the App observation and the user goal to generate appropriate actions (Wen et al., 2024a; Li et al., 2024), while multi-modality-based methods enhance this capability by incorporating GUIs for a more comprehensive understanding of the interface (Christianos et al., 2024; Wang et al., 2024b). However, these approaches struggle with long-horizon tasks that require multiple interdependent actions and a deep understanding of dynamic environments (Chae et al., 2024). For this challenge, a straightforward solution is to use real-world emulators to simulate GUI changes from user actions, enabling App agents to navigate complex scenarios and improve decision-making accuracy. However, emulators face significant drawbacks, including the safety risks from real-world interactions, such as repeatedly sending messages or making purchases. To overcome these, world models have gained attention as a more efficient alternative, not only in agents (Chae et al., 2024; Gu et al., 2024), but also in broader domains such as robotics (Li et al., 2025; Zhou et al., 2024) and self-driving (Hu et al., 2022).

### 2.2 WORLD MODEL

By observing the real world, world models can predict how the environment evolves in response to an action (LeCun, 2022; Ding et al., 2024). For instance, GameNGen (Valevski et al., 2024) predicts how a game system will respond to user actions. Notably, the ability to anticipate potential outcomes of actions has proven to be highly beneficial in informing decision-making processes (Pascanu et al., 2017; Yang et al., 2024; Schrittwieser et al., 2020; Hafner et al., 2019). Inspired by their success, world models have emerged to predict the next observation on websites. These models (Chae et al., 2024; Gu et al., 2024; Liu et al., 2023) typically take a website observation and an action as inputs to generate a textual description of the next observation. While websites provide multiple sources of information, including the actual site and their CSS or HTML source files, mobiles present a more limited context, as only the GUIs are typically accessible. Moreover, text-only descriptions such as Burns et al. (2024) often lack the precise visual details required for accurately predicting future observations, highlighting the need for a visual world model capable of generating high-fidelity future GUI images.

### 2.3 GUI GENERATION

With the rapid advancements in image generation techniques (Rombach et al., 2022; Kumari et al., 2023; Cao & Gong, 2024), previous methods have explored generating GUI directly in pixel space. For instance, layout generation methods generate the location of GUI elements (Lu et al., 2023; Zheng et al., 2023; Sobolevsky et al., 2023; Zhao et al., 2019), scene-text generation methods generate text that aligns with the visual context (Chen et al., 2024c; Zhang et al., 2024c; Chen et al., 2024b; Zeng et al., 2024), UI-diffuser (Wei et al., 2024) fine-tune a stable diffusion model to generate mobile GUIs conditioned on text prompts. For the next GUI generation conditioned on current GUI observation and a user action, it seems straightforward to resort to an image-and-text-conditioned

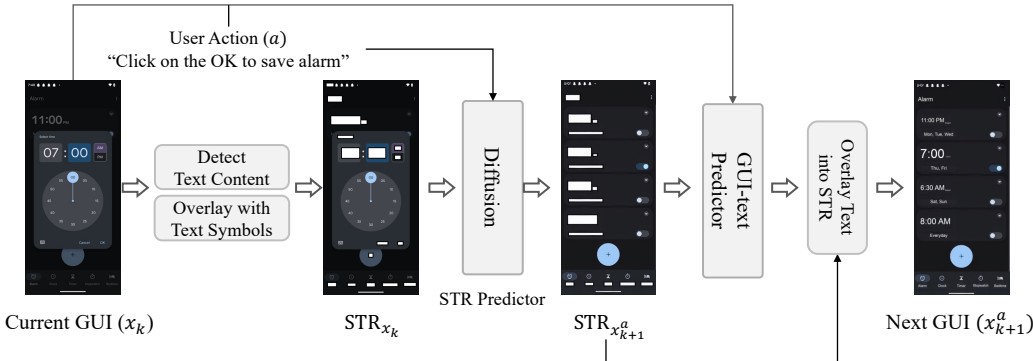

Figure 2: Framework of our ViMo. We first detect text content (actual words) in the current GUI ($x_k$) and overlay it with text symbols (rectangle-shaped placeholders with a black border and white fill), to create STR$_{x_k}$. Then STR$_{x_k}$ and the user action ($a$) are input to the STR predictor to generate the STR of the next GUI (STR$_{x_{k+1}^a}$). Next, text symbols within STR$_{x_{k+1}^a}$ are located and assigned unique ID token. Then the LLM predicts the text content corresponding to each token. Finally, the next GUI image is constructed by overlaying the predicted text into the STR.

approach (Brooks et al., 2023). However, we find that pixel-based image generation struggles with rendering text accurately, as even minor pixel prediction errors can lead to distortions, particularly for small-sized text (see Fig. 1 for examples).

In this work, we advance beyond existing approaches that generate GUI entirely (graphic and text) at the pixel level. Instead, we render graphics as image pixels and generate text as language tokens, enabling a more accurate method for GUI generation.

## 3 METHOD

In this section, we first define our setup in Subsection 3.1. Then, we introduce ViMo in Subection 3.2. Finally, we demonstrate how ViMo can be applied to enhance existing App agents in real-world scenarios (Subsection 3.3). All the prompts in this section are listed in Appendix H.1.

### 3.1 PROBLEM SETUP

In general, a GUI world model processes a given GUI Image $x_k$ at step $k$, and user action $a$, to predict the effect of $a$ on $x_k$ and simulate the next GUI. Formally, this can be expressed as:

$$x_{k+1}^a = f(x_k, a), \tag{1}$$

where $f(\cdot)$ represents the world model, and $x_{k+1}^a$ denotes the predicted next ($k$+1) GUI image after applying $a$ to $x_k$. In the following, we explain in detail of our world model.

### 3.2 VIMO: GENERATIVE VISUAL GUI WORLD MODEL

To tackle the limitation of existing methods (Wei et al., 2024; Chen et al., 2024c; Brooks et al., 2023) in generating visually plausible text for a GUI, as shown in Fig. 1, we propose ViMo, a novel generative visual GUI world model that decouples the graphic and text content generation. As shown in Fig. 2, we first detect and remove all the text in the GUI by overlaying it with a text symbol to create the Symbolic Text Representation (STR). Then a STR predictor is leveraged for determining the STR representation of the next GUI with a pixel-based diffusion process. Finally, a GUI-text predictor is proposed to generate the text content for each symbol using an LLM, followed by a handcrafted design to overplay the text into the STR image to create the next GUI. Their details are specified in the following.

### 3.2.1 STR: Symbolic Text Representation

To develop a GUI prediction model that eliminates the need to generate specific text content, we propose the Symbolic Text Representation (STR), where all the text content (actual words) within the GUI image is symbolised (overlayed) with uniform text symbols (placeholders). To be specific, we create an STR representation from a given GUI image with three steps: 1) using an OCR model (Shi et al., 2016; Qiao et al., 2020) to detect text within the GUI; 2) masking the detected text by overlaying it with a box filled with white and bordered in black; 3) we leverage an LLM to filter out static text displayed on static GUI elements and preserve it in the image, as it does not involve any semantic evolution or dynamic changes and remains unchanged as part of specific elements such as a keyboard or a clock face. Additionally, we empirically find that predicting this static text with complex spatial patterns poses significant challenges for the LLM.

Through the above process, GUI images are transformed into the STR representation, where the text content is abstracted into a text symbol, relaxing the task of generating semantic text content into predicting text symbols that indicate the location and size.

### 3.2.2 STR Predictor

Building on the powerful generative capability of diffusion-based models (Rombach et al., 2022), we introduce a STR predictor specifically trained to understand a given STR and a user action, enabling it to generate the corresponding next STR effectively. In particular, we fine-tune a pre-trained stable diffusion model (Rombach et al., 2022) to predict the next STR, conditioned on the STR of the current GUI and the user action. Given a STR representation ($\text{STR}_{x_k}$) extracted from GUI ($x_k$), the process starts with the encoding of $\text{STR}_{x_k}$ into a latent representation (Kingma & Welling, 2013): $z = \mathcal{E}(\text{STR}_{x_k})$. Gaussian noise is then added to this representation to create $z_t$ at timestep $t$. A denoising autoencoder is subsequently trained to predict the Gaussian noise in the latent representation, aiming to reverse the noise addition. The objective is defined as:

$$L = \mathbb{E}_{\mathcal{E}(\text{STR}_x), \epsilon \sim \mathcal{N}(0,I), t}\left[\|\epsilon - \epsilon_\theta(z_t, \mathcal{E}(\text{STR}_{x_k}), t, a)\|_2^2\right], \tag{2}$$

where $\epsilon_\theta$ is a U-Net (Ronneberger et al., 2015) architecture conditioned on a timestep $t$, a text prompt $a$ (action), the visual input $z_t$ and the image condition $\text{STR}_{x_k}$. To support the condition on images, we follow IP2P (Brooks et al., 2023) to add additional input channels to the first convolutional layer, concatenating the image condition $\mathcal{E}(\text{STR}_{x_k})$ with the noised latent $z_t$. After training, our STR predictor is capable of synthesising the next STR ($\text{STR}_{x_{k+1}^a}$) for $\text{STR}_{x_k}$ with action instruction $a$.

### 3.2.3 GUI-Text Predictor

Given a STR representation generated by our STR predictor, we design a GUI-text predictor to generate plausible text for the text symbols in the STR based on its graphics. Specifically, we first locate the text symbols in the STR by colour matching and boundary detection. This outputs the location of text symbols, along with their unique ID tokens assigned via enumeration, denoted as $\mathcal{T}$. Then we leverage the image processing and task understanding ability of LLM to predict the text content based on its context in STR. This process can be formulated as:

$$\text{text}_{x_{k+1}^a} = \text{LLM}(\text{STR}_{x_{k+1}^a}, x_k, a, \mathcal{T}), \tag{3}$$

where $\text{STR}_{x_{k+1}^a}$ denotes the STR representation of $x_{k+1}^a$. $\text{text}_{x_{k+1}^a}$ contains the predicted text content for each text symbol, associated with its ID token. This design ensures flexible and accurate text generation tailored to the predicted GUI STRs as the context. Finally, we overlay each text content ($\text{text}_{x_{k+1}^a}$) to $\text{STR}_{x_{k+1}^a}$ to reconstruct the predicted GUI image ($x_{k+1}^a$). To be specific, text symbols are replaced with the corresponding text based on coordinates, with dynamic styling determined by the symbol's size and surrounding colours. More details are provided in Appendix A.1.

### 3.3 ViMo Enhanced App Agent

Motivated by that App agents usually face limitations in long-horizon planning to make optimal decisions on action selection (Chae et al., 2024), we leverage the proposed ViMo to enhance the decision-making of App agents.

---

**Algorithm 1** Enhancing App Agent with Generative Visual GUI World Model

---

**Input:** Current GUI Observation $x_k$, A goal $g$, A visual world model ViMo, A selection model $S(\cdot)$.
**Output:** Action to be applied on $x_k$ to achieve $g$.
Generate action options $\mathcal{A}$ with $n$ actions $\{a^i\}$ (Eq. (4)).
**for** $i = 1$ **to** $n$ **do**
    Leverage ViMo to synthesise the next GUI observation conditioned on $a^i$ and $x_k$, denoted as $x_{k+1}^{a^i}$ (Eq. (5)).
**end for**
Use $S(\cdot)$ to identify the optimal action with predicted observation (Eq. (6)).

---

Table 1: GUI quality evaluation. $s_{gc}$ indicates the GUI consistency, $s_{ia}$ instructional accuracy, $s_{ar}$ action readiness score and $s_h$ the harmonic average between the 3 metrics. $\Delta s_h$ is the relative performance gains of our ViMo over other methods. IP2P* denotes finetuing of IP2P on our dataset.

| Method | Automatic Metric | | | | | User Study | | | | |
|---|---|---|---|---|---|---|---|---|---|---|
| | $s_{gc}$ | $s_{ia}$ | $s_{ar}$ | $s_h$ | $\Delta s_h$ | $s_{gc}$ | $s_{ia}$ | $s_{ar}$ | $s_h$ | $\Delta s_h$ |
| HTML-vision | 0.70 | **85.77** | 62.79 | 0.72 | 5.39% | 0.31 | 11.32 | 9.01 | 0.23 | 282.61% |
| IP2P* | **0.74** | 63.57 | 70.15 | 0.69 | 10.20% | 0.82 | 58.92 | 52.81 | 0.63 | 39.68% |
| UI-diffuser | 0.60 | 39.61 | 38.75 | 0.44 | 71.82% | 0.36 | 14.32 | 8.56 | 0.27 | 225.93% |
| ViMo (Ours) | **0.74** | 75.39 | **78.68** | **0.76** | - | **0.89** | **91.12** | **84.71** | **0.88** | - |

To be specific, we break down the process into three steps: action option generation, action outcome synthesis, and action selection. In the first step, the App agent generates $n$ action options, as follows:
$$\mathcal{A} = \text{Agent}(x_k, g), \qquad (4)$$
where $\mathcal{A} = \{a^1, a^2, \cdots, a^n\}$ denotes the action option set, $x_k$ is the current GUI image at step $k$, and $g$ is the given user goal. With these action options, our world model ViMo is leveraged to synthesise the outcome (next GUI) of these actions as follows:
$$x_{k+1}^{a^i} = \textbf{ViMo}(x_k, a^i), \qquad (5)$$
where $x_{k+1}^{a^i}$ denotes the synthesised next GUI of applying action $a^i$ on $x_k$. Finally, each action $a^i$ and its corresponding predicted outcome $x_{k+1}^{a^i}$ are fed into an LLM-based selection model, which identifies the optimal action based on the generated GUIs. This process can be formulated as:
$$a_{se} = S\left(\{(a^i, x_{k+1}^{a^i})\}_{i=1}^n\right), \qquad (6)$$
where $a_{se}$ denotes the selected action, and $S(\cdot)$ is the selection model. This procedure is outlined in Algorithm 1. By predicting the next GUI, ViMo provides the agent with the potential outcome of an action, enabling it to make more informed decisions. We build the selection model to identify the best action in two steps. First, we query an LLM to evaluate all the action candidates, providing a judgment—either *valid* or *invalid*—and a confidence score for each action. Second, we query the LLM again to select the best action from the two highest-scoring actions. This process is motivated by our observation that, in over $70\%$ of tasks, the difference between the top two scores is equal to or less than $0.1$, indicating that both are likely optimal. By explicitly prompting the LLM to compare the top candidates, we go beyond coarse scoring and enable more detailed decision-making.

## 4 EXPERIMENTS

In this section, we begin by summarising our proposed STR dataset discussed in Subsection 3.2.1. Next, we tested the core capability of our ViMo, focusing on its GUI generation ability. Building on this, we demonstrated how the powerful GUI generation capability of ViMo can enhance the decision-making of App agents. Then, we studied our effectiveness in real-world App navigation tasks. Finally, we carried out the ablation study to validate the effectiveness of our model design. Specific setups and experiment details are elaborated in subsequent sections. Unless explicitly stated otherwise, GPT-4o (Hurst et al., 2024) was employed as the default LLM in the following sections.

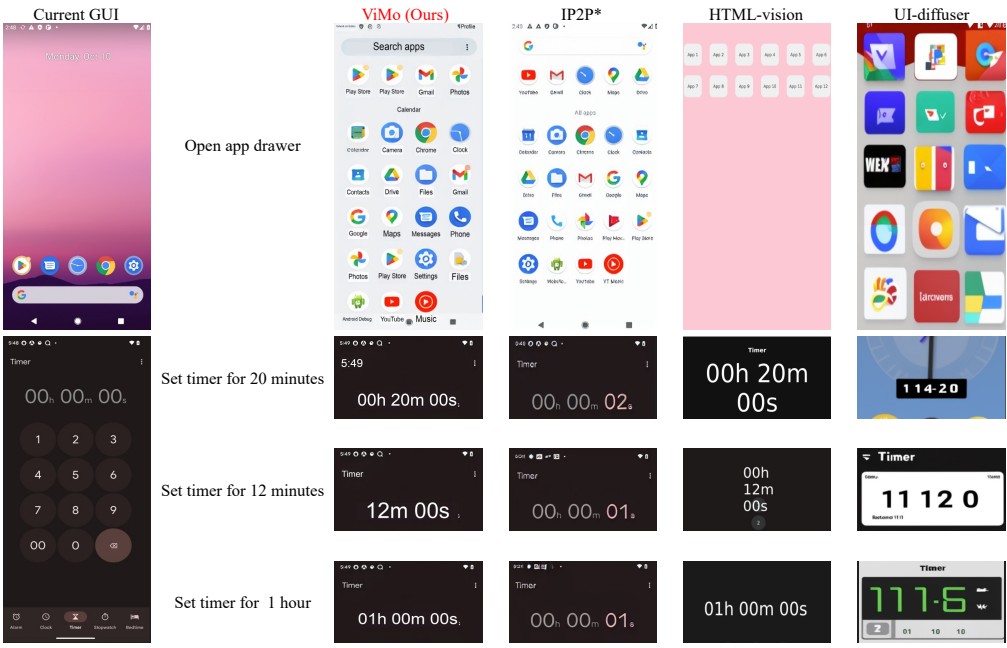

Figure 3: GUI generation comparison in graphic generation (Top) and text generation (Bottom).

## 4.1 DATASET SUMMARISATION

Our STR dataset was constructed using data from two widely recognised and large-scale sources: Android Control (Li et al., 2024) and Android in the Wild (AITW) (Rawles et al., 2024b). From these sources, we respectively sampled 12 and 7 Apps, selecting those with rich data samples while filtering out noise. Android Control provides two types of user actions: 1) *action commands*: predefined actions (e.g., click, scroll) accompanied by specific parameters such as coordinates (x, y); 2) *action instructions*: actions described in natural languages, such as "click the plus icon". We used action instructions as conditions for our world model as this approach was more concrete and better utilised the pre-trained model in understanding natural language. For AITW, action commands were converted into action instructions using GPT-4o (Hurst et al., 2024). In total, we collected 19 Apps with 3,550 episodes, 23,620 images and 18,450 actions. To ensure both time-efficient and cost-efficient experiments, we followed prior App agent (Rawles et al., 2024b) on partial split evaluation. Specifically, we randomly sampled 57 episodes across 19 distinct Apps. Details of dataset collection, split summarisation, and full-split experiments are provided in Appendices A.3 and B.

## 4.2 WORLD MODEL ABILITY

We evaluated the GUI generation capability of ViMo by GUI quality evaluation. We included IP2P (Brooks et al., 2023) and UI-diffuser (Wei et al., 2024), both originally designed for image editing and GUI generation. We fine-tuned IP2P on our dataset to generate everything of the GUI, including the text content and the graphic, denoted as IP2P*. We also leveraged an LLM to predict App observations in an HTML format, which were rendered into images, denoted as HTML-vision.

We leveraged 3 evaluation metrics: The *GUI consistency score ($s_{gc}$)* assessed the visual similarity between the ground truth and the generated next GUI; *Instructional accuracy score ($s_{ia}$)* determined whether the generated GUI adheres to the user action; *Action readiness score ($s_{ar}$)* evaluated whether the generated GUI retains valid elements essential for subsequent actions required to achieve the user goal. Both automatic evaluation and user studies were conducted. For the automatic evaluation, we used DINO (Caron et al., 2021) as the visual encoder to compute $s_{gc}$, and an LLM to evaluate $s_{ia}$ and $s_{ar}$. For the user study, we invited 70 voluntary participants to complete questionnaires based on 80 GUI samples, generated by all 4 compared methods. For each sample, participants were asked

Table 2: Performance comparison of app agents on step accuracy. Apps are categorised into "Leisure", "Work", and "System".

| Agent Type | App Agent | Leisure | Work | System | Overall |
|---|---|---|---|---|---|
| Language-Based | ER (Li et al., 2024) | 31.76 | 46.15 | 34.13 | 34.50 |
| | AutoDroid (Wen et al., 2024a) | 35.81 | 46.15 | 31.75 | 35.46 |
| | T3A (Rawles et al., 2024a) | 41.22 | 51.28 | 42.86 | 43.13 |
| | T3A + ViMo (**Ours**) | 50.00 | **58.97** | 45.24 | 49.20 |
| Multi-Modality-Based | APP-Agent (Zhang et al., 2023) | 43.24 | 51.28 | 39.68 | 42.81 |
| | Mobile-Agent-v2 (Wang et al., 2024a) | 43.92 | 53.85 | 39.68 | 43.45 |
| | M3A (Rawles et al., 2024a) | 46.62 | 51.28 | 43.65 | 46.01 |
| | M3A + ViMo (**Ours**) | **53.38** | 53.85 | **45.24** | **50.16** |

Table 3: Compare World Models.

| Modality | World Model | Step Acc. |
|---|---|---|
| w/o WM | w/o WM | 46.01 |
| Language | Change-text | 47.28 |
| | HTML-text | 46.65 |
| Vision | HTML-vision | 48.89 |
| | UI-diffuser | 47.60 |
| | IP2P* | 48.56 |
| | **ViMo (Ours)** | **50.16** |

Table 4: Zero-shot Evaluation.

| App Agent | LLM | Step Acc. |
|---|---|---|
| SeeAct | GPT-4-Turbo | 33.9 |
| M3A | GPT-4-Turbo | 42.1 |
| ER | Gemini 1.5 Pro | 24.4 |
| T3A | Gemini-2.0-Flash | 41.4 |
| T3A+ViMo | Gemini-2.0-Flash | 46.8 |
| M3A | Gemini-2.0-Flash | 44.2 |
| M3A+ViMo | Gemini-2.0-Flash | **47.6** |

three questions, one each for evaluating $s_{gc}$, $s_{ia}$, and $s_{ar}$. Details of the prompts for the LLM-based evaluations and the full instructions for the user study are described in Appendix A.4.

As shown in Table 1, ViMo achieved the highest score on the harmonic average of the three automatic metrics, surpassing other methods with an average relative performance improvement of 29.14%. The results of the user study were consistent with the automatic evaluations, where ViMo demonstrated the best performance. Notably, HTML-vision and UI-diffuser performed significantly worse in human evaluation compared to their scores in the LLM-based assessment. This discrepancy likely arose because human evaluators perceived the outputs of these methods as visually unrealistic or functionally incoherent, leading to lower subjective scores in $s_{gc}$, $s_{ia}$ and $s_h$.

Qualitative comparisons are presented in Fig. 3, under two scenarios: GUI graphic changes (Top) and text generation (Bottom, cropped for space efficiency). Experiments revealed that while the HTML-vision method exhibited greater flexibility in responding to user actions (as shown in the bottom examples), it failed to produce concrete details necessary for future actions (top). Conversely, IP2P* generated plausible GUI graphics but lacked flexibility in text content generation (also reflected by $s_{gc}$ and $s_{ia}$ in Table 1). This trade-off highlighted the superior balance of ViMo.

### 4.3 WORLD MODEL ENHANCED APP AGENT

This section demonstrates that: 1) ViMo enhanced the performance of App agents in decision-making; 2) ViMo outperformed other world models in enabling App agents to make more accurate decisions.

**Comparison with App Agents.** In this experiment, we collected 6 LLM-based App agents, which included three language-based methods: ER, AutoDroid, and T3A, as well as three multi-modality-based methods: APP-Agent, Mobile-Agent-v2, and M3A. We applied our ViMo into M3A and T3A following the process in Subsection 3.3. Moreover, we followed the previous works (Rawles et al., 2024b; Li et al., 2024) to use the step accuracy (the number of correct actions divided by the number of overall actions) to quantify the model performance. To provide more detailed results, we categorised the Apps into three groups: "Leisure", "Work" and "System". Table 2 demonstrates that ViMo was beneficial to the App agent, achieving a relative performance gain of 9.01% for M3A and 14.07% for T3A. These findings highlighted the effectiveness of our proposed world model in providing App agents with enhanced decision-making capability. Additional information about the categorisation and experiments with more App agents are provided in the Appendix A.4.

Table 5: Online Evaluation on Android World (Rawles et al. (2024a)).

| App Agent | LLM | Task Acc. |
|---|---|---|
| SeeAct | GPT-4-Turbo | 15.50 |
| M3A | GPT-4-Turbo | 25.40 |
| M3A | Gemini-1.5-Pro | 22.80 |
| T3A | GPT-4-Turbo | 30.60 |
| T3A | Gemini-1.5-Pro | 19.40 |
| T3A | Gemini-2.0-Flash | 33.19 |
| T3A + ViMo | Gemini-2.0-Flash | **40.95** |

Table 6: Ablations on preserving static text and using action instructions, measured by step accuracy.

| Static Text | Action Instr. | App Agent | |
|---|---|---|---|
| | | T3A | M3A |
| N/A | N/A | 43.13 | 46.01 |
| ✓ | – | 42.81 | 45.05 |
| – | ✓ | 47.28 | 48.88 |
| ✓ | ✓ | **49.20** | **50.16** |

**Comparison with World Models.** We evaluated the ability of ViMo to enhance App agent decision-making by comparing it against existing world models. In addition to vision-based world models discussed in Subsection 4.2, we also incorporated two language-based world models (Gu et al., 2024; Chae et al., 2024), utilising Change-text to generate textual descriptions capturing differences between consecutive observations and HTML-text to predict App observations in an HTML format. Then, we applied each world model to M3A App agents. Table 3, together with Appendix Table 9, illustrates that vision-based methods consistently outperform language-based world models, thereby reinforcing our motivation for developing visual GUI world models. Moreover, our approach achieves superior performance over existing world models, underscoring its effectiveness and advantage.

## 4.4 REAL-WORLD APPLICATIONS

**Practical Deployment.** ViMo was designed to be lightweight and easily deployable. The minimum requirement for deployment is a GPU with 16 GB of memory. Moreover, ViMo was implemented as a plug-and-play API that required only a single function call, making integration straightforward. Inference time on V100 GPU is 8 seconds on a STR image generation and 30 seconds on GUI-text prediction. We collected and compared the inference time with existing methods in the Appendix C.

**Generalisation to New Apps.** Generalisation is a crucial capability for real-world applications. To assess the generalisation performance of our method on new Apps that were unseen during training, we conducted a zero-shot evaluation using data from the Android Control dataset (Li et al., 2024), explicitly excluding Apps encountered during training. As shown in Table 4, ViMo substantially outperformed the baseline and achieved 47.6%, underscoring its robustness and adaptability to novel App environments. Additional visualisations of unseen scenarios are provided in the Appendix F.

**Online Evaluation.** To further demonstrate the effectiveness of ViMo in realistic App navigation scenarios, we conducted an online evaluation using the AndroidWorld dataset (Rawles et al., 2024a), which comprises 116 distinct navigation tasks. Performance was measured using the task success rate (Task Acc.). As illustrated in Table 5, ViMo achieved a notable improvement of 7.76% over the baseline method, highlighting its effectiveness and reliability in real-world settings.

## 4.5 ABLATION STUDY

In this section, we ablated on three key components of ViMo: 1) preserving static text within the image to simplify the text generation task; 2) using action instructions instead of action commands as the conditioning input for ViMo; and 3) varying the number of iterations, where each iteration corresponds to one roll-out step into the future during GUI prediction.

Firstly, for the challenge of predicting static text from specific GUI elements, such as keyboard, number pad or clock face, which typically did not involve text

Table 7: Ablation on the number of iterations.

| Method | Iterations | Step Acc. (%) |
|---|---|---|
| T3A | N/A | 39.94 |
| T3A+ViMo | 1 | 46.06 |
| | 2 | **46.65** |
| | 3 | 45.05 |

changes and exhibited complex spatial patterns, we retained static text within the image (Subsection 3.2.1). This approach eliminated the need for the LLM to generate such static text while generating

in pixels instead. Secondly, we proposed conditioning STR prediction on action instruction rather than action commands (Subsection 4.1). Ablation results are presented in Table 6, where "Static Text" indicates whether static text was retained in the images, and "Action Instr." denotes whether natural language instructions ("√") or abstract action commands ("-") were used as conditioning input to ViMo. The first row indicates the baseline where ViMo was not applied. The table shows that both components contributed significantly to performance improvements across the two App agents, highlighting their critical roles in enabling ViMo to generate high-quality GUIs. Visual comparison examples are provided in the Appendix F for further illustration.

Our ViMo predicted future GUI observations, which could be recursively fed back as input to simulate further into the future. In this ablation study, we varied the iteration number to evaluate how extended roll-outs impact prediction accuracy. We took Gemini-2.0-Flash (Hassabis & Kavukcuoglu, 2024) as the LLM in this study. As shown in Table 7, performing two iterations yielded the highest accuracy. However, this also led to increased computational cost. Therefore, we selected one step as a practical trade-off between performance and efficiency. We also observed a slight decline in performance at iteration 3 relative to iterations 1 and 2, indicating that extending the prediction horizon did not necessarily improve agent behaviour. This was likely due to that longer horizons introduced not only additional foresight but also a greater accumulation of prediction errors, whose detrimental effect could outweigh the potential benefits. Further analysis of error accumulation, user examples of ViMo with App agent, and comparisons with various world models are provided in Appendices B and F.

## 5 CONCLUSION

In this work, we introduced ViMo, a novel generative visual GUI world model designed to predict App observations in a visual modality, providing a more realistic and concrete approach compared to contemporary language-based models. To address the unique challenges of GUI generation, ViMo was equipped with the STR representation to simplify text content generation to text location prediction by overlaying text content with placeholders and delegating content generation to LLM. This innovation ensured high visual fidelity and avoided artefacts like distorted or blurred text. Through extensive experiments, we demonstrated that ViMo generated both visually plausible and functionally effective GUIs. Notably, ViMo boosted step-wise action prediction accuracy by a relative performance gain of 14.07%, underscoring its potential to enhance decision-making of App agents. Furthermore, real-world experiments demonstrated the strong generalisation ability of ViMo to unseen Apps, along with its robust performance in online navigation tasks under real-time environment interaction. Together with its superiority over language-based world models, these results highlighted the value of ViMo in advancing GUI world modelling in visual modality.

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

In this Appendix, we first provide detailed explanations, including prompts related to our methods, descriptions of our STR dataset, and evaluation details. Then, we present additional experimental results. Finally, we present additional visualisations of our proposed ViMo for GUI generation.

# A  EXPERIMENTAL DETAILS

## A.1  GUI-TEXT PREDICTOR

This subsection elaborates on the design and functionality of the GUI-text predictor, summarising its key components and providing a detailed explanation of its underlying processes.

Given a STR prediction, the GUI-text predictor starts by locating the text symbols. To be specific, we first detect black borders by identifying black pixels in the BGR colour space, generating a binary mask that indicates whether a pixel is black or not. A pixel is classified as black if its BGR values fall within the range [0,0,0] to [50,50,50]. Next, we identify rectangular regions within this mask by computing the ratio of the actual contour area to its corresponding bounding rectangle area. If this ratio exceeds 0.8, the region is considered a valid rectangle, allowing us to extract rectangles with black borders. For these detected regions, we further analyse their internal colour distribution to determine whether they contain the desired white colour. Specifically, we define white pixels as those with BGR values within the range [200,200,200] to [255,255,255]. If more than 50% of the pixels within a region fall within this range, the region is classified as a text symbol. Thus, the locations of text symbols are extracted, and we assign a unique identifier (ID) to each symbol through enumeration.

Building on this, we take as inputs the current GUI image $x_k$, an action $a$ to be applied to this image, the predicted STR ( $\text{STR}_{x_{k+1}^a}$ ), the location and unique ID token of the text symbols in the STR $\mathcal{T}$. Then we leverage an LLM to predict the text content for each text symbol. The process begins with preprocessing the STR by overlaying the ID token for each text symbol to the corresponding position in the STR image, resulting in a modified representation denoted as $\text{STR}_{k+1}^{ID}$. Next, we prompt an LLM to identify which text symbols will remain unchanged after the action $a$ (see the prompt in Subsection H.1). These symbols are determined to not be affected by the action and have content identical to the previous GUI $x_k$. Based on the resulting ID list, we retrieve the corresponding pixels from the previous GUI $x_k$ based on their location and update the STR representation. The updated image is still referred to as $\text{STR}_{k+1}^{ID}$ for simplicity.

Subsequently, the LLM is prompted to determine the semantic role of each text symbol by analysing its context (see the prompt in Subsection H.2). This semantic information, combined with $\text{STR}_{k+1}^{ID}$, is then used to predict the exact text content of each symbol (see the prompt in Subsection H.3).

Finally, to overlay a symbol with its actual text content, we perform the following steps: 1) For a given text symbol's location and corresponding text, the average background colour is computed by the average colour of the area on the edge of text symbol's coordinates; 2) The text colour is set to either white or black to ensure optimal contrast with the background colour, for better visibility; 3) The font size is calculated as the maximum size that allows the text to fit entirely within the boundaries of the text symbol, ensuring optimal use of space and readability.

## A.2  ACTION SELECTION

In practice, our selection model, described in Section 3.3, identifies the best action in two steps. First, we query an LLM to evaluate all the action options, providing a judgment—either *valid* or *invalid*—and a confidence score for each action (see the prompt in Subsection H.4). These judgments are transformed into scores: if an action is judged *valid*, its score equals the confidence; if judged *invalid*, its score is the confidence multiplied by $-1$. This scoring reflects that higher confidence in a *valid* action yields a higher score, while higher confidence in an *invalid* action results in a lower (negative) score. Second, we query the LLM again to select the best action from the two highest-scoring actions (see the prompt in Subsection H.5). This step is motivated by our observation that, in over 70% of tasks, the difference between the top two scores is equal to or less than 0.1, indicating that both are likely optimal. By allowing the LLM to choose between them, we refine the selection beyond simply picking the action with the highest score.

Table 8: Summarisation of our STR dataset.

| Split | App | Episode | Image | Instrucion |
|-------|-----|---------|-------|------------|
| Train | 19 | 2853 | 19010 | 14852 |
| Val | 19 | 349 | 2290 | 1774 |
| Test | 19 | 348 | 2320 | 1824 |
| All | 19 | 3550 | 23620 | 18450 |

Table 9: Decision optimisation comparisons on APP agent performance. Apps are categorised into "Leisure", "Work", and "System".

| App Agent | World Model Modality | World Model | Leisure | Work | System | Overall |
|-----------|---------------------|-------------|---------|------|--------|---------|
| T3A | w/o world model | w/o world model | 41.22 | 51.28 | 42.86 | 43.13 |
| | Langugae | Change-text | 49.32 | 51.28 | 42.06 | 46.65 |
| | | HTML-text | 47.30 | 48.72 | 43.65 | 46.01 |
| | Vision | HTML-vision | 50.68 | 53.85 | 43.65 | 48.24 |
| | | UI-diffuser | 48.65 | 53.85 | 43.65 | 47.28 |
| | | IP2P | 48.65 | 53.85 | **45.24** | 47.92 |
| | | **ViMo (Ours)** | 50.00 | 58.97 | **45.24** | 49.20 |
| APP-Agnet | w/o world model | w/o world model | 43.24 | 51.28 | 39.68 | 42.81 |
| | Langugae | Change-text | 45.96 | 56.41 | **45.24** | 46.96 |
| | | HTML-text | 44.59 | 56.41 | **45.24** | 46.33 |
| | Vision | HTML-vision | 47.97 | 56.41 | 46.03 | 48.24 |
| | | UI-diffuser | 47.30 | 56.41 | 44.44 | 47.28 |
| | | IP2P | 47.30 | 58.97 | **45.24** | 47.92 |
| | | **ViMo (Ours)** | 50.68 | 58.97 | 43.65 | 48.89 |
| Mobile-Agent-v2 | w/o world model | w/o world model | 43.92 | 53.85 | 39.68 | 43.45 |
| | Langugae | Change-text | 47.30 | **66.67** | 41.27 | 47.28 |
| | | HTML-text | 47.30 | **66.67** | 38.89 | 46.33 |
| | Vision | HTML-vision | 50.00 | **66.67** | 41.27 | 48.56 |
| | | UI-diffuser | 49.32 | 61.54 | 41.27 | 47.60 |
| | | IP2P | 46.62 | **66.67** | **45.24** | 48.56 |
| | | **ViMo (Ours)** | 50.00 | **66.67** | 44.44 | 49.84 |
| M3A | w/o world model | w/o world model | 46.62 | 51.28 | 43.65 | 46.01 |
| | Langugae | Change-text | 51.35 | 51.28 | 41.27 | 47.28 |
| | | HTML-text | 50.68 | 51.28 | 40.48 | 46.65 |
| | Vision | HTML-vision | 52.03 | 48.72 | **45.24** | 48.89 |
| | | UI-diffuser | 50.00 | 48.72 | 44.44 | 47.60 |
| | | IP2P | 52.03 | 48.72 | 44.44 | 48.56 |
| | | **ViMo (Ours)** | **53.38** | 53.85 | **45.24** | **50.16** |

## A.3 DATA COLLECTION

To ensure the quality and diversity of data samples for each App, while minimising noise, we collected App information from both Android Control (Li et al., 2024) and Android in the Wild dataset (AITW) (Rawles et al., 2024b) datasets. To be specific, out of 15,274 episodes in the Android Control, only 5,697 episodes include the "open_app" action. From these episodes, we extracted their "app_name", identifying 758 unique applications. However, only 13 of these Apps had more than 50 samples. To enrich the dataset, we manually collected additional samples for these 13 Apps from the rest of the dataset. For AITW, we extracted App names by using the package name listed under the "current activity" field. After filtering out the noisy, 11 valid Apps remained. By combining the overlapping applications from both datasets, we obtained a total of 19 unique Apps. We split our dataset into "Train", "Validation" and "Test" splits, and we summarise our dataset under each split in Table 8.

Furthermore, we converted action commands into action instructions for AITW with specific prompts in Subsection H.6. We use Paddleocr (Shi et al., 2016) for STR generation.

## A.4 EVALUATION

**World Model Ability.** For the results under automatic metrics presented in Table 1, we prompt LLM for the instructional accuracy score $s_{ia}$ and action readiness score $s_{ar}$, as shown in Subsection H.7 and Subsection H.8 respectively. A generation is considered successful if "success" appears under "Status" for $s_{ia}$ and "yes" under "ready for action" for $s_{ar}$. For the user study, we collected 80

Table 10: Trajectory synthesis evaluation. "T+L" denotes the accuracy of the whole trajectory with length L.

| World Model | T+1 | T+2 | T+3 | T+4 |
|---|---|---|---|---|
| w/o world model | 22.81 | 14.04 | 7.02 | 0 |
| Change-text | 52.63 | 26.32 | 10.53 | 5.26 |
| HTML-text | 38.60 | 14.04 | 12.28 | 7.02 |
| HTML-vision | 43.86 | 19.30 | 10.53 | 10.53 |
| UI-diffuser | 52.63 | 29.82 | 12.28 | 5.26 |
| IP2P* | 56.14 | 21.05 | 10.53 | 7.02 |
| **ViMo** (Ours) | **57.89** | **36.84** | **14.03** | **12.28** |

Table 11: Evaluation on randomness by running the experiment 3 times (r1-r3) on our sampled test split. "All" denotes the evaluation of the full test split. $s_{gc}$, $s_{ia}$ and $s_{ar}$ are the metrics same with Table 1. $s_h$ denotes their harmonic score. STD denote the standard deviation from r1 to r3.

| World Model | $s_{gc}$ | $s_{ia}$ | $s_{ar}$ | $s_h$ |
|---|---|---|---|---|
| r1 | 0.7421 | 75.08 | 78.29 | 0.7582 |
| r2 | 0.7323 | 75.63 | 77.64 | 0.7546 |
| r3 | 0.7423 | 75.39 | 78.68 | 0.7605 |
| STD | 0.0057 | 0.23 | 0.42 | 0.0025 |
| ALL | 0.7389 | 75.37 | 78.20 | 0.7578 |

generated samples—20 from each of the four world models. We then asked 70 participants to answer three questions on each sample designed to reflect the $s_{ia}$, $s_{gc}$ and $s_{ar}$ scores, as detailed in Subsection H.9. For the $s_{gc}$, participants are asked to rate on a scale from 1 to 5. These scores were then normalised to the [0,1] range in Table 1.

**World Model Enhanced App Agent.** In Table 2, we categorised APPs based on their primary functions into three groups: **Leisures**, **Work**, and **System**. The **Leisure** category includes APPs commonly used for relaxation and entertainment, such as *Decathlon*, *eBay*, *Flipkart*, *Amazon*, *Adidas*, *Kitchen Stories*, *Booking.com*, *YouTube*, and *Vimeo*. The **Work** category comprises APPs typically associated with professional or productivity-related activities, including *Gmail*, *Drive*, and *Chrome*. Lastly, the *System* category encompasses APPs pre-installed in the Android operating system, such as *com.android.contacts*, *com.google.android.dialer*, *com.google.android.googlequicksearchbox*, *com.android.settings*, *com.google.android.APPs.maps*, and *com.android.vending*.

**Ablation on Iteration Numbers.** ViMo predicts future GUI observations, which can be recursively fed back as input to simulate further into the future. Taking the generative GUI as the current GUI, an agent was prompted to generate the action instructions based on the user goal (see the prompt in Subsection H.10). Then the action instruction and the GUI were fed into ViMo to generate the next GUI. In this study, we defined the iteration number as the number of times ViMo was called. We only use the final output as the signals during the candidate action selection phase, guiding the final selection among potential actions.

## B ADDITIONAL EXPERIMENTAL RESULTS

**Comparison with World Models.** Table 3 compares our ViMo with existing world models under M3A App agent. To further highlight our superiority, Table 9 presents additional results of ViMo applied to T3A, APP-Agent, and Mobile-Agent-V2. The experimental results indicate that vision-based methods consistently outperform their language-based counterparts, thereby substantiating our motivation for developing visual GUI world models. Moreover, our approach achieves superior performance over existing world models, underscoring its effectiveness and advantage.

**Generation Error Analysis.** As discussed in Subsec. 4.5, our method ViMo can iteratively generate future GUIs. However, as the number of iterations increases, the accumulated error also grows. In

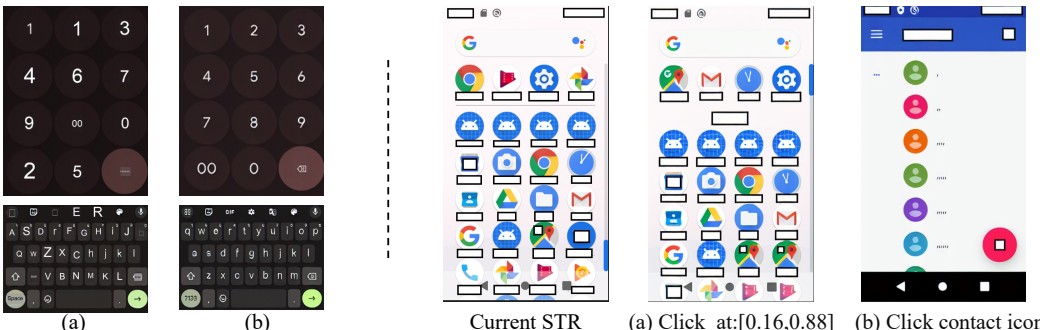

Figure 4: Qualitative ablation studies. Left: Static text generation. (a) Generating static text via an LLM; (b) Preserving the original text in the image by rendering it as image pixels. Right: STR generation under two input formats—(a) action command and (b) action instruction.

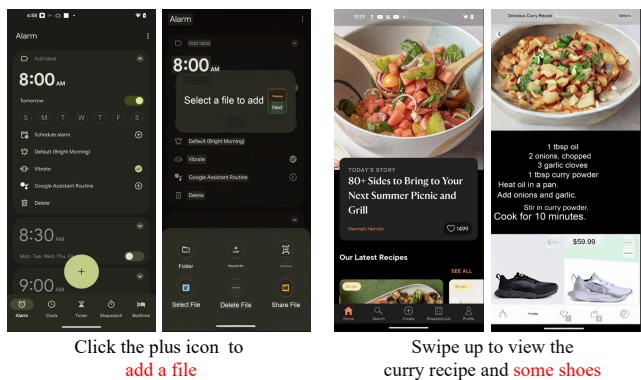

Figure 5: GUI generation conditioned on a novel combination of current GUI observation and user action.

addition to the evidence presented in Table 7, we conduct further experiments to analyse this iteration error and compare our approach with existing world models.

To this end, we design a trajectory synthesis evaluation to assess how well the GUIs generated by ViMo align with those observations in real-world environments over longer iterations. In this setup, the generated GUI is leveraged as the input to an App agent to generate the subsequent action, with higher-quality trajectories indicating a GUI more aligned with the real-world environment. Specifically, the GUIs generated by ViMo serve as the observation input for the App agent, which generates actions aimed at achieving the user's goal. These output actions are then evaluated to reflect whether the GUI representations offer concrete and reliable information for action prediction. This process is repeated for L steps, and we calculate the success rate of the entire L-step trajectory.

We employ an LLM as a judge to assess the alignment between the agent's simulated actions and the ground truth actions within a given trajectory. Specifically, an agent was prompted to generate the action instructions based on the given GUI and the user goal (see the prompt in Subsection H.10) and the LLM evaluated whether the simulated action lead to the same outcome as the ground truth action (see the prompt in Subsection H.11), a "yes" of the "Status" is calculated as a match.

As shown in Table 10, we compared ViMo against both visual- and language-based world models and demonstrated that while performance decreases across all world models with more iterations, our model significantly outperformed the other methods by providing more accurate and reliable information. This was reflected in higher trajectory prediction accuracy, underscoring the ability of our model to generate GUIs that aligned with the real-world environment.

Table 12: Comparison of inference runtime and step accuracy across different models. "Execution Time" denotes the per-module execution time for the vision and text components. The end-to-end time includes model execution, initialisation, loading, and communication overhead. "–" indicates that the corresponding module is not applicable.

| Model | Execution Time (Vision / Text) | End-to-End Time | Step Accuracy (%) |
|---|---|---|---|
| Baseline (T3A) | – / – | 4 min | 43.13 |
| Change-text | – / 5s | 5 s | 46.64 |
| IP2P* | 8s / – | 1.5 min | 47.92 |
| ViMo (Ours) | 8s / 30s | 2 min | **49.20** |

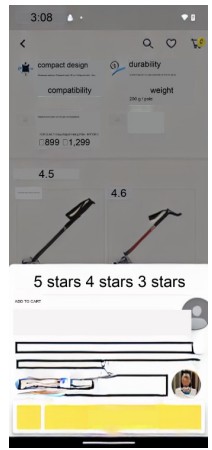 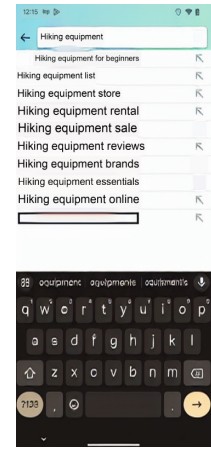 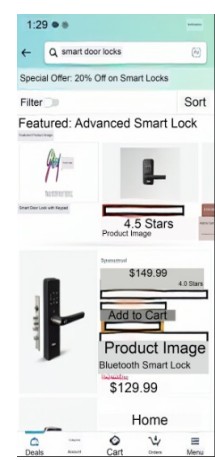 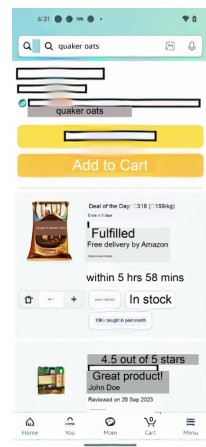

Figure 6: False examples where the text symbols are incorrectly represented, making them unrecognisable to indicate the location of text.

**Randomness Study and Evaluation on Full Test Split.**    ViMo involves random factors, particularly from the use of LLMs. The LLM is prompted to generate plausible textual content, and in some cases, multiple reasonable options can be produced. For example, in Fig. 3, it shows "5:49 "on the top left corner for "set timer for 20 minutes" command and shows "Timer" for "set timer for 12 minutes", both are plausible and valid in the given context. However, the key functional element, the timer itself, is consistent with the user instructions in both cases. To evaluate their influence, we conducted the experiment three times, as summarised in Table 11 (r1-r3). The results demonstrate that the randomness does not significantly impact the performance or consistency of our method. Additionally, we focused on a randomly selected subset of examples for evaluation, with results from the full test set also included to illustrate that the observed differences are minor, as shown in Table 11 (compare ALL to r1-r3). We consider the subset results to provide an accurate and reliable approximation for our analysis.

**Qualitative Ablation Analysis.**    In addition to the quantitative ablation results presented in Table 6, we also provide qualitative comparisons. Fig. 4 (left) illustrates the challenges faced by the LLM in predicting static text under complex spatial layouts. Fig. 4 (right) displays the STR generation of the same user intent but with different action types. It demonstrated that models learned with action commands failed to predict STR that aligns with the user's intent, whereas action instructions offered a more concrete description, enabling the model to better capture the intent.

**Qualitative Generalisation Study.**    We studied the generalisation of ViMo in Fig. 5 by providing user actions that were not typically encountered within the App's standard context. For example, in the Clock App, a user action to "add a file" generated a Drive-style file selection window while retaining the Clock interface. Similarly, in the Kitchen Store App, ViMo can generate content corresponding to the action. These results emphasised ViMo's generalisation ability facing novel combinations of App observations and user actions.

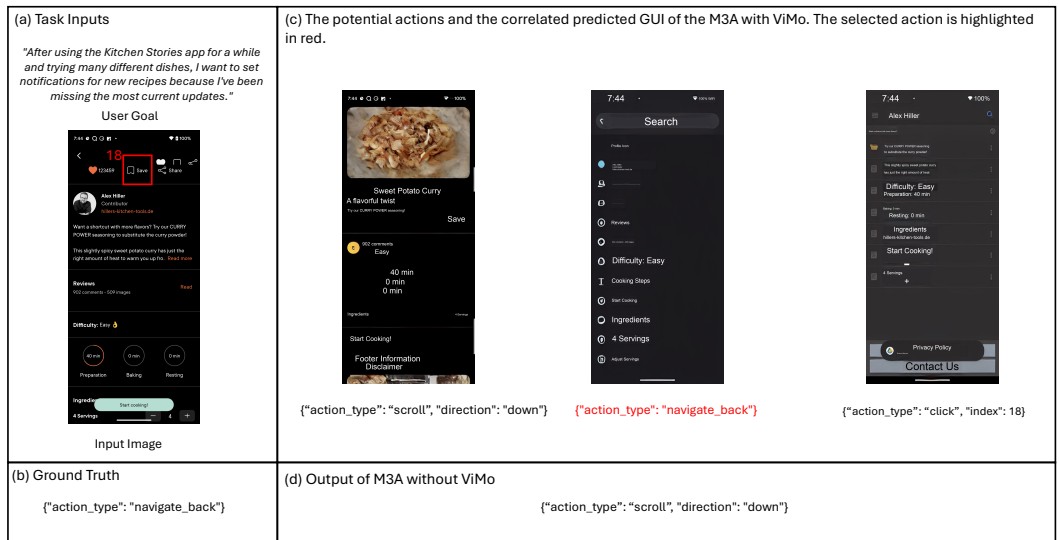

Figure 7: Example of how ViMo helps the App agent to select the correct action.

## C PRACTICAL DEPLOYMENT

In this section, we report the computational efficiency of our method to demonstrate its practicality in real-world applications. The minimum hardware requirement is a GPU with 16 GB of memory. As shown in Table 12, with a V100 GPU, STR image generation (Vision) takes approximately 8 seconds, and GUI-text prediction (Text) takes around 30 seconds. These runtimes are practical and we have demonstrated an online inference setting in Table 5.

Although ViMo introduces additional latency, this is an intentional design choice that is closely aligned with our objective of improving the reliability of long-horizon decision making. For example, when the planning horizon is extended to 4 steps, our ViMo pipeline achieves 2.33× higher accuracy (12.28% vs. 5.26% in Table 10) compared to Change-text and 1.75× higher to IP2P*. We view the ability to construct a visual world model that can accurately predict the consequences of actions as a critical foundation for building robust GUI agents. While this design involves a higher computational cost, we believe this trade-off is justified by the substantial gains in reliability and planning accuracy, especially when the horizon is longer.

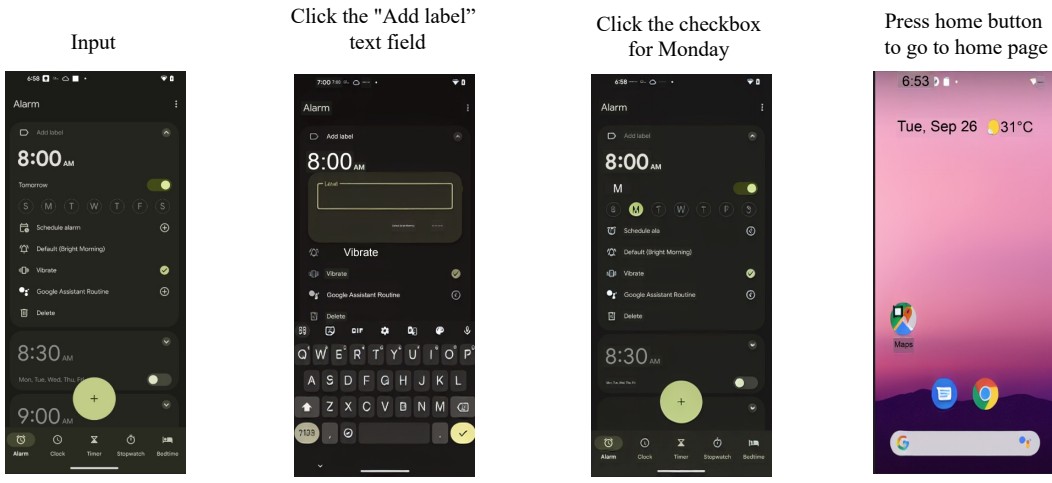

Figure 8: Visualisation of ViMo in generating GUIs given a single current GUI paired with different actions.

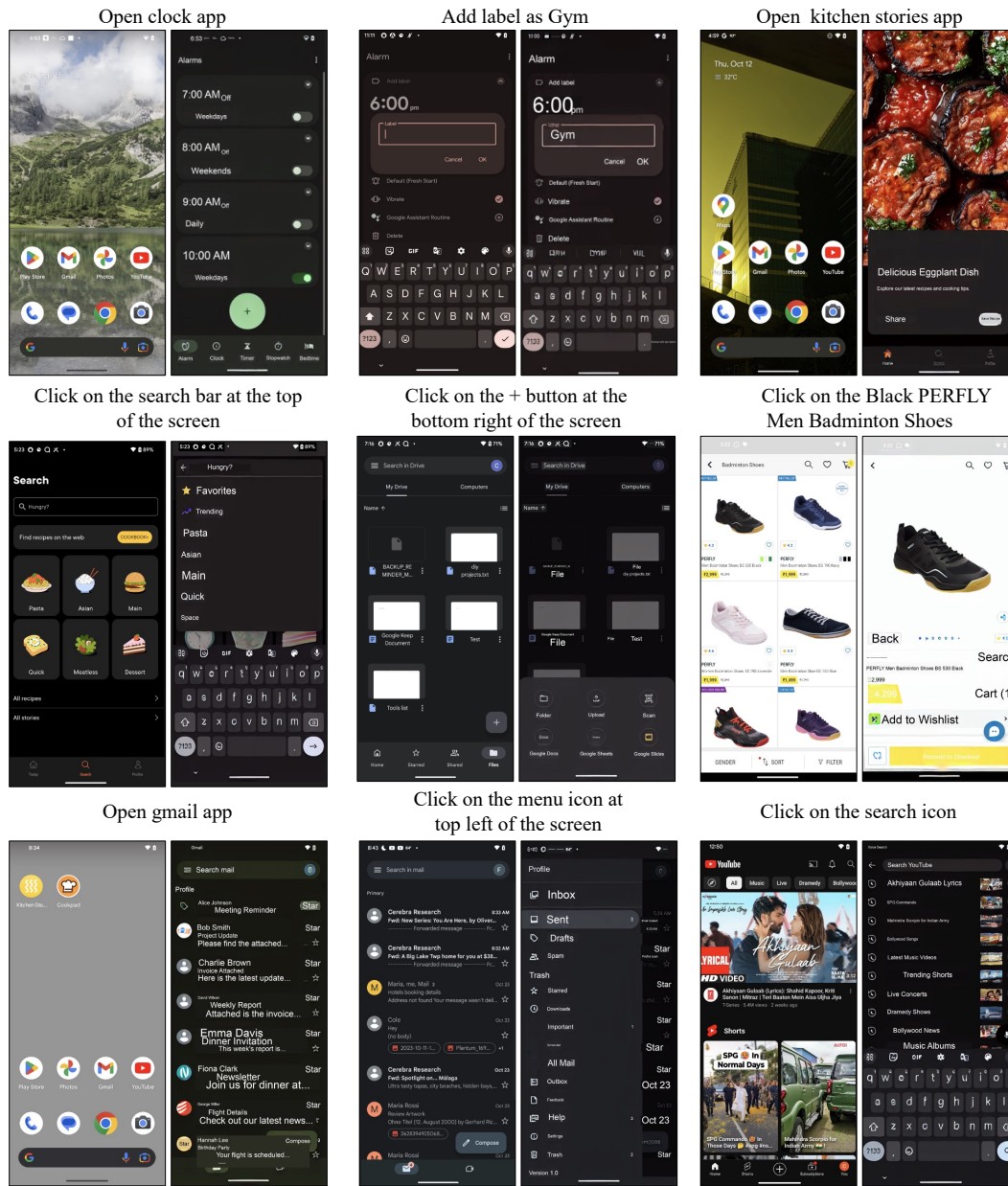

Figure 9: Visualisation of ViMo in generating the next GUI. For each example, the action is displayed at the top, with the current GUI shown on the left and the generated GUI on the right.

# D    ADDITIONAL DISCUSSION

## D.1    USER PRIVACY PRESERVATION

We emphasise that our method neither collects sensitive user data nor has the potential to leak such information.

For GUI graphic generation, the diffusion model is trained on our STR representation (GUI graphic), in which all textual content is removed from the GUI. The model is trained solely to predict visual and layout information. As a result, no sensitive or personally identifiable information is included in the training data or generated by the model.

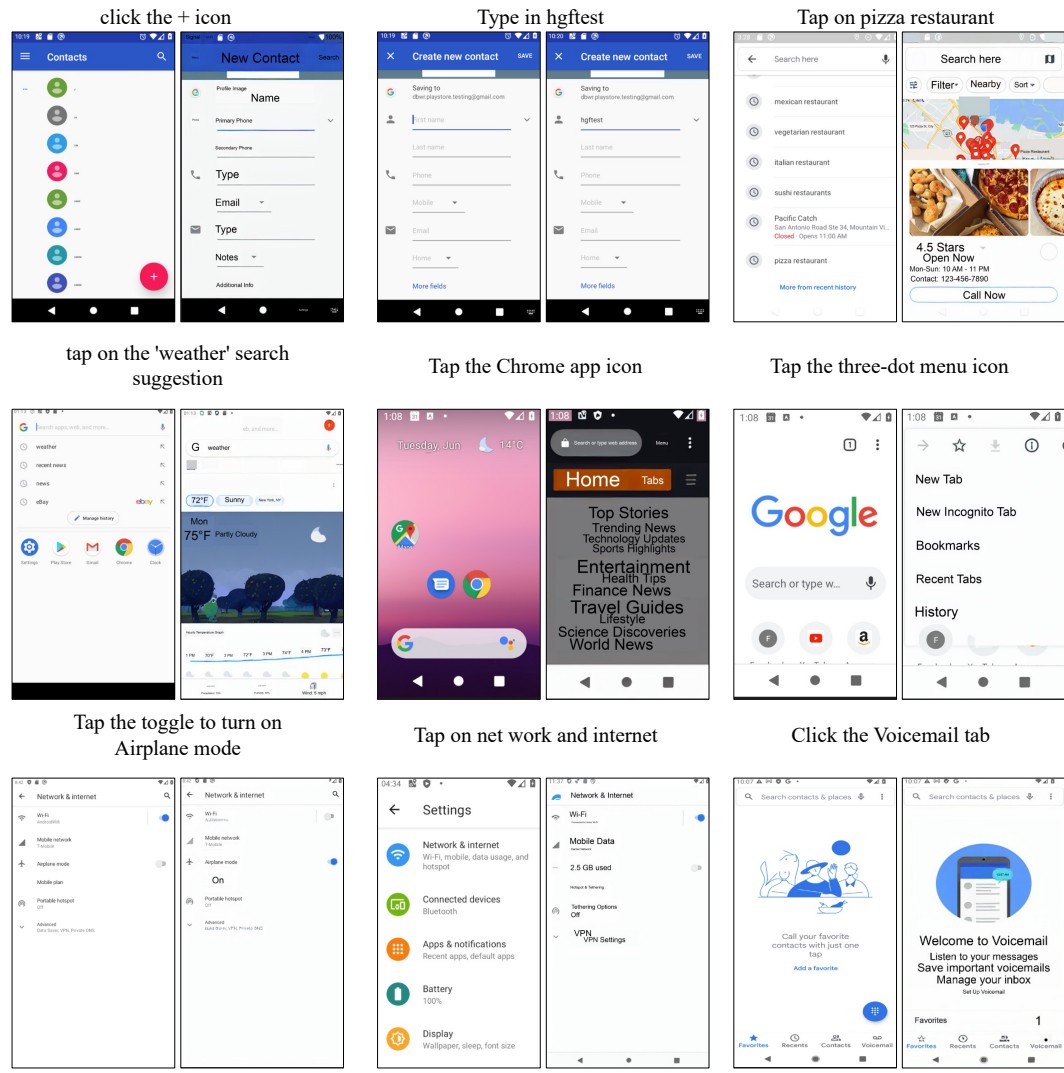

Figure 10: Visualisation of ViMo in generating the next GUI. For each example, the action is displayed at the top, with the current GUI shown on the left and the generated GUI on the right.

For text rendering, no model fine-tuning is performed. At test time only, an LLM (such as ChatGPT) is prompted to generate textual content based on contextual information such as the previous GUI state, the user action, and the user goal. Sensitive user information that is not displayed on the screen is never seen or accessed. Sensitive information that may appear on screen (e.g., contact lists or account details) is not collected, stored, or reused for training or any other purpose.

Therefore, no private user information is collected, stored, or leaked during either the training or inference stages.

## D.2    FAIRNESS COMPARISONS

To evaluate whether the visual modality provides more precise information than the language modality for app agent tasks, we conduct controlled comparisons under a non-finetuned setting. As shown in Tables 3, 9, and 10, our two visual baselines (HTML-vision and UI-diffuser), which are not fine-tuned on our task, consistently outperform the language-based methods (Change-text and HTML-text), which are also not fine-tuned. This result reflects the inherent advantage of visual representations for modelling fine-grained GUI details.

To evaluate the necessity of our ViMo design, we also finetune the vision-based baseline IP2P on our task using the same training data as ViMo. This ensures a fair comparison and further verifies that the observed improvements stem from our model architecture rather than advantages in the training data.

## E  LIMITATION AND FUTURE WORKS

Fig. 6 illustrates failure cases where text symbols are not represented as our rectangle-shaped placeholders with a black border and white fill, making them unrecognisable as text symbols. Improving the representation of text symbols remains a potential direction for future work.

Beyond visual quality, computational efficiency remains a practical limitation. Improving the efficiency of robust GUI agents is a critical research direction for enabling real-world deployment.

## F  ADDITIONAL VISUALISATION

Fig. 7 demonstrates how ViMo helps the M3A App agent make better action decisions. Fig. 8 showcases results generated from a single current GUI paired with different actions, further highlighting the versatility of our approach. Diverse visualisations are presented in Fig. 9 and Fig. 10. These examples illustrate how our method effectively generates the next GUI based on the given action and current GUI observation, showcasing its ability to produce visually coherent and contextually accurate GUI simulations.

## G  STATEMENT ON LLM USAGE

We disclose that large language model (LLM) tools were used solely for language refinement of the manuscript, including improving grammar and polishing phrasing. LLMs were not used to generate scientific content, research ideas, experiment designs, data, analyses, or code. All suggestions and modifications from these tools were made under the direct supervision and final approval of the authors, and all authors are fully aware of and consent to this usage.

## H  PROMPTS

### H.1  PROMPT TO DECIDE THE TEXT SYMBOLS TO REMAIN UNCHANGED AFTER THE ACTION

```
You are a professional UI/UX analyst and your goal is to compare the two UI screenshots
and return their overlapping layout.
### Inputs:
1.  **Current Screenshot**:  The current mobile UI as an image.
2.  **Next UI Layout Screenshot**:
- An image of the next mobile UI layout with all text replaced by white boxes.
- Each box has a unique red ID label.
3.  Use action:  a user action described by language
Next UI Layout Screenshot is a result of a user action on the current screenshot, but
the text elements are masked.
Please help me identify those layouts that are located in the same position, so I can
predict their text directly from the current screenshot.
Usually, the system bar information should be included.  Exclude elements from the
result if:
The content (text) changes as a result of the user action, even if the element exists
in both screenshots.
Please be very very cautious about putting an ID on the list, which means you are very
very confident with this task.  if you are unsure about some elements, please ignore
them and do not put them on the list.
### Output the list of existing elements :  Return the results in the following JSON
format:  ['id1','id2',...]
### Notes:
- Ensure the detected elements appear in both UI screenshots, which means their
surrounding context is the same.
- Ensure identify those elements that their text will change by the user action and
exclude them from your response.
- Ensure identify those elements that share a similar context layout, but their absolute
are not the same, and them from your response.
- Ensure only reply in pure JSON format, with no placeholders or comments.
```

## H.2   PROMPT TO DETERMINE THE SEMANTIC ROLE OF EACH TEXT SYMBOL

```
You are a professional UI/UX analyst assigned to structure and analyse the semantics of
mobile UI screenshots.
Your goal is to segment the UI and annotate box elements in a way that enhances
understanding of their roles and relationships within the interface.  Inputs:
- Current Screenshot:  A visual representation of the mobile UI.
- Next UI layout screenshot:  A visual representation of the next UI layout with all the
text masked with a white box.  Each box has an ID number on it in red colour.  - User
Action:  An action put on the current UI will result in the next UI.
- Box locations:  a list of box locations to better help you to locate the boxes in the
format of 'id': id, 'Location':[x1,y1, width, height].  ID indicates their ID number
in the UI screenshot.
- UI_size:  the width/height of the input images.  They are the same size.  The image
you received might be resized.  Please scale it back for the locations.
Task:
Structure the boxes in the Next UI layout screenshot with semantics based on the visual
input by following these steps:
1, Divide the UI into Semantic Windows Group the UI into functional sections with a
specific name (e.g., "Header Windows," "Time Selector Panel").
2.   Structure Text Elements in Each Semantic Window.
- Assign box elements to windows based on logical, visual relationships or semantic
roles.
- For every element, structure output as :
**id:  corresponding box retrieved from the box list and the Next UI layout screenshot.
**Role:  A brief explanation of the role of this box.  You should consider their [x1,y1]
to indicate their location, [w,h] to indicate their size to decide the role.  It is
important to consider the context for the role prediction.  The role should be in
detail to distinguish it from other items in the same category.
Output Format:  { "Window Name":    "Category Name": [ "id":id, "Role":  "Role" ,
"id":id, "Role":  "Role" , ... ], "Category Name": [ "id":id, "Role":  "Role" , ...
]  , ... }
Key Guidelines:
- Ensure to retrieve id from the given screenshot and box list.
- Avoid duplicating or omitting IDs.
- Every box element in the box location list must be included in the structured output.
- Ensure there is no additional formatting, code blocks or placeholders in your
response; return only a clean JSON without any comments.
```

## H.3   PROMPT TO PREDICT THE EXACT TEXT CONTENT FOR EACH SYMBOL

```
Task:  Plan the content for the next UI screen based on the provided inputs and
instructions.
Inputs:
Current Screenshot:  A visual representation of the mobile UI.
Next UI layout screenshot:  A visual representation of the next UI layout with all
light yellow boxes indicating a text place.  Each box has an ID number on it.
User Instruction:  A specific action or command that transitions the current UI to the
next UI state.
Semantics for the masks in Next UI screenshot:  A structured map.
Goal:
Predict the content (text) for each masked area in the next UI layout screenshot based
on the following steps:
Map Affected Elements to the Next UI.
Align the affected elements with the yellow box coordinates on the next UI.
Predict the text for each yellow box based on the user instruction and the context of
the current UI.
If you can not find any information about the text, predict a plausible text based on
its context.
Ensure to use the semantics to help you understand the layout and predict the text.  If
you think the semantics is wrong, please modify it in your
Output:
Return the predictions in JSON format with the structure: {"Window Name ":  "Category
Name ":  [ "id ":  id, "text ":  "text", "role ":  "role" , "id ":  "id", "text":
"text", "role":  "role "  ], , ... }
Ensure to predict text based on the context.
Do not include any special characters.
Ensure there is no additional formatting, code blocks or placeholders in your response;
return only a clean JSON without any comments.
```

## H.4 Prompt to evaluate actions with a confidence score

```
You are an agent who can operate an Android phone on behalf of a user.  When given
a user request, you will try to complete it step by step.  At each step, a list of
descriptions for most UI elements on the current screen will be given to you (each
element can be specified by an index), together with a history of what you have done
in previous steps.  Based on these pieces of information and the goal, you must choose
to perform one of the actions in the following list (action description followed by the
JSON format) by outputting the action in the correct JSON format:  action options from
the dataset
The overall user goal/request is:  {goal}
Here is a history of what you have done so far:{history} This is the action you picked
in the latest step:  {action}, whose semantic description is:  {sum}
Your goal is to judge **whether the action you picked in the latest step is on the
right track to the successful execution of the overall user goal/request**.
You will be given the screenshots before and after you perform the action
- The first screenshot corresponds to the UI state before you performed the action.
- The second screenshot corresponds to the UI state after you performed the action.
Also here is the list of detailed information for some UI elements in the before
screenshot:  {before_elements}
Note that, the "after" screenshot is generated by the agent's world model.  As such,
it may not faithfully represent the real UI. For instance:  Some UI elements in
the simulated "after" screenshot may not exist in a real UI. Your evaluation should
consider the reliability of the UI predictions.  If the "after" screenshot contains
unreasonable elements, this likely indicates a failure.
Now provide your judgment on the selected action in JSON format.  Your response must
include:
Reason:  A detailed explanation of why the action is valid or invalid.
Judgment:  Your judgment must be either "valid" or "invalid".
Confidence:  A confidence score between 0.0 and 1.0, reflects how likely your judgment
is correct.
You must follow this structure exactly:
{Reason:  ..., Judgement:  "valid" or "invalid", Confidence:  ...}
Your Answer:
```

## H.5 Prompt to select the optimal actions among two highest-scoring actions

```
You are an agent who can operate an Android phone on behalf of a user.  When given
a user request, you will try to complete it step by step.  At each step, a list of
descriptions for most UI elements on the current screen will be given to you (each
element can be specified by an index), together with a history of what you have done
in previous steps.  Based on these pieces of information and the goal, you must choose
to perform one of the actions in the following list (action description followed by the
JSON format) by outputting the action in the correct JSON format action options from
the dataset
The overall user goal/request is:  {goal}
Here is a history of what you have done so far:{history}
Here is a list of descriptions for some UI elements on the current
screen:{before_elements}
Here are two candidate actions:
Action 1:  {action_0}, described semantically as {sum_0}.  The rationale for this
action is:  {act_re_0}
Action 2:  {action_1}, described semantically as {sum_1}.  The rationale for this
action is:  {act_re_1}
Hints for making your decision:  {GUIDANCE}
- Both "more options" buttons and scrolling actions may reveal new content.  Evaluate
which is more suitable for the goal.
- Consider the history of previous actions.  If prior steps involved repeated "scroll
down" actions, it is more likely that "scroll down" is the correct next step.
- If the user goal involves viewing reviews or similar tasks and the current screen
already displays such content, "scroll down" may reveal more information.
Your task is to choose the best action from the two provided.
Now, provide your judgment in JSON format with the following structure:
Reason:  A detailed explanation of your choice, considering the hints above.
Choice:  Action 1 or Action 2.
Your output must exactly match this format:
{Reason:  ..., Choice:  Action 1 or Action 2}
```

## H.6 PROMPT TO CONVERT ACTION COMMANDS INTO ACTION INSTRUCTIONS

```
You are a professional UI/UX analyst specializing in identifying the semantics of dual
point actions between mobile UI screenshots.
Inputs:
Current Screenshot:  A visual representation of the mobile UI.
Next Screenshot:  A visual representation of the NEXT mobile UI.
Goal:  A user intent on this Mobile interface.
touch_xy:  the x,y coordinates for the touch point, as a percentage of the image
dimensions.
lift_xy:  the x,y coordinates for the lift point, as a percentage of the image
dimensions.
Your task is to analyse these elements describe the precise user action in plain
language and return your answer in plain string (e.g., "click the + icon", "scroll
up").
If the two screenshots are identical, please return an empty string as "".
If the Next Screenshot does not seem to be one step away from the Current Screenshot,
return an empty string as "".  One step means only one interaction with the cell phone.
Ensure there is no additional formatting, code blocks or placeholders in your response;
return only a clean string without any comments
```

## H.7 PROMPT FOR INSTRUCTIONAL ACCURACY SCORE ($s_{ia}$)

```
You are an expert in evaluating the performance of a mobile emulator.  The mobile
emulator is designed to navigate the UI change based on human instruction.
Inputs:
Current UI Screenshot:  The present state of the cellphone's user interface.
Next UI Screenshot:  The mobile emulator generated UI indicating the next state of the
cellphone's user interface based on human instruction.
Human instruction:  The action applied on the current UI screenshot.
Your goal is to determine whether the mobile emulator successfully predicts the next UI
image with current information and layout based on the current UI and the user action.
*IMPORTANT*
Format your response into a JSON map as shown below:
{
"Thoughts":  <your thoughts and reasoning process>,
"Status":  "success" or "failure",
}
```

## H.8 PROMPT FOR ACTION READINESS ACCURACY SCORE ($s_{ar}$)

```
You are an expert in evaluating the performance of a mobile emulator.  The mobile
emulator is designed to navigate the UI change based on human instruction.
Inputs:
UI Screenshot:  The mobile emulator generated UI indicating the state of the
cellphone's user interface.
User intent:  The user goal to achieve.
Next action:  the action will be applied to this UI.
Your goal is to determine whether the next action is validated on the UI Screenshot.
Please also indicate if it is still in the right App according to the goal.
*IMPORTANT* Format your response into a JSON map as shown below:
{
"Thoughts":  <your thoughts and reasoning process>,
"In the right App":  "yes" or "no"
"ready for action":  "yes" or "yes",
}
```

## H.9 INSTRUCTIONS FOR USER STUDY

The following prompt provides the instructions for the user study. An example screenshot is shown in Fig. 11.

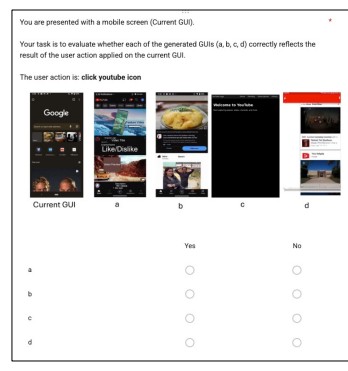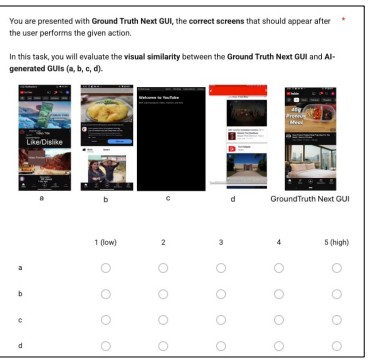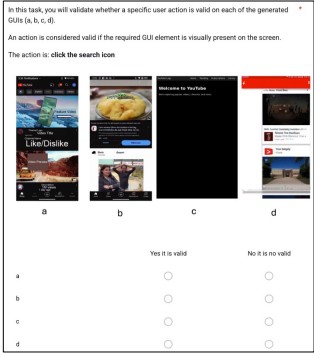

Figure 11: Screenshot of user study example.

```
Question 1:  You are presented with a mobile screen (Current GUI).
Your task is to evaluate whether the generated GUI correctly reflects the result of the
user action applied on the current GUI. Answer "Yes" or "No" to each sample.
Question 2:
You are presented with Ground Truth Next GUI, the correct screens that should appear
after the user performs the given action.
In this task, you will evaluate the visual similarity between the Ground Truth Next GUI
and AI-generated GUI, scoring from 1-5.
Question 3:
In this task, you will validate whether a specific user action is valid on the
generated GUI.
An action is considered valid if the required GUI element is visually presented on the
screen.  Answer "Yes" or "No" to each sample.
```

### H.10 PROMPT TO GENERATE THE ACTION INSTRUCTION BASED ON THE GIVEN GUI AND THE USER GOAL

```
You are an autonomous intelligent agent tasked with navigating a cell phone to
accomplish specific tasks.  You will be provided with the following information:
1.  Initial UI screenshot:  A visual representation of the initial state of the cell
phone's interface.
2.  User Objective:  This is the task you are trying to complete.
3.  Previous Action:  An action sequence performed on the initial UI.
4, Current UI states:  A visual representation of the current state of the cell phone's
interface, generated by a simulated environment.
The initial image is the screenshot before actually performing all the previous
actions.
The current cell phone UI is generated by applying previous actions on the initial
screenshot.
Your Task:  Please predict a single next step action to complete the given task based
on current vision states.
To be successful, it is very important to follow the following rules:
1.  You should only issue one action that is valid based on the current UI states.
2.  You should only issue one action at a time.  Avoid issuing multiple actions like
"do A and do B".
3.  Generate the action in plain text.  For example, Scroll down to set the minute as
15.
4.  Issue "Stop." if you think the action is already completed.  Ensure you only return
the action, not other formats, comments or placeholders
```

### H.11 PROMPT TO EVALUATES WHETHER THE SIMULATED ACTION LEADS TO THE SAME OUTCOME AS THE GROUND TRUTH ACTION

```
You are an expert in evaluating the performance of a cell phone navigation agent.  The
agent is designed to help a human user navigate a cellphone to complete a task.
Inputs:
Current UI Screenshot:  The present state of the cellphone's user interface.
User Intent:  The goal the human user aims to achieve.
Action History:  The sequence of actions taken so far for you to track the progress.
Agent Simulated Action:  The action suggested by the agent to achieve the user's
intent.
Ground Truth Action:  The correct action is needed to achieve the user's intent.
Your goal is to determine whether the agent's simulated action leads to the same
outcome as the ground truth action.
Additionally, if the simulated action does not exactly match the ground truth action
but is still progressing toward the correct outcome to achieve user intent, indicating
that the action is "on the right track."
*IMPORTANT*
Format your response into a JSON map as shown below:
{
"Thoughts":  <your thoughts and reasoning process>,
"Status":  "success" or "failure",
"On the right track to success":  "yes" or "no"
}
```

