# OpenReview forum: "ViMo: A Generative Visual GUI World Model for App Agents"
_ICLR.cc/2026/Conference — ICLR 2026 Poster_

### Official Review · Reviewer_udhG · 2025-10-31

**Soundness:** 3
**Presentation:** 3
**Contribution:** 3
**Rating:** 8
**Confidence:** 4

**Summary:**

This paper presents ViMo, a generative visual world model for mobile GUI agents. It addresses the failure of text-only world models to capture essential visual details and the failure of general image models to render text legibly. The paper's core idea is to decouple GUI generation: a diffusion-based *STR Predictor* generates the non-text graphical layout, and an LLM-based *GUI-text Predictor* generates the text content for the predicted layout. ViMo is then used as a lookahead mechanism to help a base agent select the best action from a list of candidates, leading to improved decision-making accuracy.

**Strengths:**

- A Novel and Necessary Model. The idea of a visual world model for GUIs is a logical and necessary next step for the field. The authors are one of the first to propose a concrete and workable architecture for this.
- The proposed STR is a good workaround to decompose the problem, allowing the diffusion model and the LLM to handle the sub-tasks they are respectively good at (spatial graphics vs. semantic text).
- Clear Downstream Task Improvement. The paper doesn't just evaluate image quality; it demonstrates a clear, practical benefit. Using ViMo to "preview" actions improves the underlying agent's step accuracy by a significant margin. The online task success rate also improves by an absolute 7.76%

**Weaknesses:**

- Complexity in architecture. The framework is a complex multi-stage pipeline (OCR -> STR creation -> Diffusion prediction -> Symbol detection -> LLM text prediction -> Rendering). A failure at any one of these steps (e.g., the initial OCR misses text, the diffusion model generates a malformed placeholder, the symbol detector fails) could cause the entire prediction to fail. Can the authors elaborate more on how to effectively build an end-to-end world model in the future? What are the fundamental challenges, and is it just data? It'll also be nice to see any initial effort or blueprint for that kind of world models.
- With 20K images, ViMo is already with pretty good capability. Do the authors have any data-centric scaling experiments or ideas if we can have more samples by simply scaling the environment and explore? Also, what would be the challenges to train a more universal digital world model that can also predict on high-res desktop envs?

**Questions:**

Please see weaknesses.

---

> ### Author Response · Authors · 2025-11-20
>
> We thank the reviewer for the positive and encouraging feedback on our work and for recognising the importance of developing a visual world model for GUI agents. Below, we provide detailed responses to the concerns raised.
>
> **Comment 1**:
> Can the authors elaborate more on how to effectively build an end-to-end world model in the future? What are the fundamental challenges, and is it just data? It'll also be nice to see any initial effort or blueprint for those kinds of world models.
>
> **Response 1**:
>
> We appreciate the reviewer's insightful comments.
>
> In our view, the core challenge in building a truly end-to-end world model for mobile GUIs lies in jointly modelling GUI graphics, textual content, and the underlying action-conditioned reasoning within a coherent latent space. While ViMo's current modules capture different aspects of this problem, the bottleneck is not only the scale of data but also the lack of multimodal (image + text) and cross-task (generation + reasoning) architectures that can unify these components seamlessly.
>
> Looking forward, an effective blueprint for such a world model would extend ViMo's design principle of separating GUI graphic generation and textual content generation, while integrating them within a unified generation framework. Such a model would generate GUI images, textual content, and intermediate reasoning steps within a single unified decoding process. A key open question is how to integrate parallel image generation with autoregressive text generation, as their fundamentally different generation mechanisms do not naturally align.
>
> We hope our discussion provides a clear starting blueprint for future work in this direction.
>
> **Comment 2**:
> With 20K images, ViMo is already with pretty good capability. Do the authors have any data-centric scaling experiments or ideas if we can have more samples by simply scaling the environment and explore?
>
> **Response 2**:
>
> ViMo's strong capability and generalisation largely stem from two factors:
>
> (1) Instead of generating GUIs from scratch, the model focuses on learning GUI transitions, which is a more structured and learnable problem. Novel examples were presented in Fig.5.
>
> (2) The use of an LLM backbone, which provides strong generalisation and generation abilities.
>
> Regarding data-centric scaling, simply expanding the environment and collecting more trajectories is indeed a promising direction. However, this also introduces several challenges. A central open problem is how to efficiently balance exploration and redundancy: naively increasing trajectories often leads to large amounts of repetitive screens, while the model benefits most from diverse and informative GUI transitions or novel action-GUI combinations. Developing exploration strategies that maximise diversity while reducing redundancy is therefore an important next step.
>
> We believe that with greater computational resources, broader environment coverage, and more targeted trajectory collection, scaling the dataset would continue to improve the model’s capabilities.
>
> **Comment 3**:
> what would be the challenges to train a more universal digital world model that can also predict on high-res desktop envs?
>
> **Response 3**:
>
> Thanks to the reviewer for this question. We summarise the challenges into three aspects:
>
> (1)
> Desktop environments typically have much higher resolutions than mobile screens. Directly encoding and generating such high-resolution images greatly increases memory and computation requirements, making end-to-end training far more challenging.
>
> (2)
> Unlike mobile GUIs with relatively stable layouts, desktop interfaces feature overlapping windows, draggable panels, toolbars, resizable components, and a much larger and more flexible action space. Modelling these highly dynamic and hierarchical structures within a unified latent representation is therefore substantially more difficult.
>
> (3)
> In desktop environments, usually only a small region of the screen changes for each action, whereas mobile GUIs have more screen transitions. A PC-targeted digital world model would therefore need a mechanism to detect which regions have changed and need to be updated locally. Such a module would avoid unnecessary full-image generation and make learning more efficient.

---

> > ### Author Response · Authors · 2025-11-27
> >
> > Dear Reviewer udhG,
> >
> > We would like to express our sincere gratitude for your positive and encouraging feedback. We hope that our clarifications and additional explanations are clear and helpful.
> >
> > Best regards

---

### Official Review · Reviewer_Wy6A · 2025-10-31

**Soundness:** 3
**Presentation:** 4
**Contribution:** 3
**Rating:** 6
**Confidence:** 4

**Summary:**

**Summary**

This paper introduces **ViMo** (Visual GUI World Model), a novel **generative world model** designed for **App agents** that interact with mobile graphical user interfaces (GUIs). Current App agents often struggle with **long-horizon planning**—they cannot accurately predict how future screens will look after performing an action. Prior “world models” attempt to simulate such outcomes but typically rely on **text-only descriptions**, which fail to capture visual layouts and pixel-level details essential for GUI reasoning.

To overcome these limitations, ViMo formulates the problem as **visual GUI prediction**: given a current screen (x_k) and a user action (a), it generates the **next GUI image** (x_{k+1}^a = f(x_k, a)). The model’s core innovation is the **Symbolic Text Representation (STR)**, a hybrid representation that decouples **graphic** and **textual** content generation. STR replaces all text on a GUI with rectangle-shaped placeholders (“text symbols”), allowing ViMo to first predict the overall layout and visual structure (using a **diffusion-based STR predictor**) and then fill in textual content (via a **LLM-based GUI-text predictor**). This design mitigates pixel-level rendering errors that commonly distort text in diffusion models.

ViMo is evaluated in three major settings:

1. **World-model ability:** ViMo achieves superior GUI generation quality compared with baselines such as IP2P*, HTML-vision, and UI-diffuser, measured by GUI consistency ((s_{gc})), instructional accuracy ((s_{ia})), and action readiness ((s_{ar})) under both automatic metrics and user studies.
2. **Agent-enhanced decision making:** When integrated into existing App agents (e.g., T3A, M3A), ViMo improves step-wise action prediction accuracy by up to **14.07%** and raises online task completion rates by **7.76%**.
3. **Real-world deployment and generalization:** ViMo runs efficiently (≤16 GB GPU, plug-and-play API) and generalizes to unseen Apps in zero-shot evaluations, achieving 47.6% step accuracy.

**Contributions:**

* Proposes **ViMo**, the **first generative visual GUI world model** capable of predicting future App observations as realistic images.
* Introduces **Symbolic Text Representation (STR)** to separate layout prediction from text generation, ensuring high visual fidelity and readable on-screen text.
* Demonstrates through extensive experiments that ViMo significantly enhances both **world-model quality** and **App-agent decision-making performance**, outperforming prior language- and vision-based baselines across multiple benchmarks.

**Strengths:**

The paper proposes a **novel and well-motivated idea** that effectively addresses the weakness of existing generative models in handling text within GUI scenes. The introduction of **Symbolic Text Representation (STR)** is both elegant and practical, enabling clear separation between visual layout generation and text rendering.

Experiments are **comprehensive and convincing**, demonstrating strong improvements in both GUI generation quality and downstream App-agent performance. The results show that ViMo not only produces more realistic and legible interfaces but also enhances action prediction and task success rates.

Overall, the work is **original**, **technically solid**, and **highly significant**, offering a new perspective on world modeling for digital interfaces with clear implications for multimodal agents.

**Weaknesses:**

While the paper’s idea is strong, the **experimental setup lags behind the current state of the field**. Most evaluations rely on relatively **outdated models and benchmarks**, such as older App-agent frameworks and base models that no longer reflect the best-performing systems. Recent models (e.g., **UITars**, **Qwen3-VL**, and other 2025-era multimodal agents) have reported around **60% step accuracy on AndroidWorld**, significantly higher than the baselines used in this paper. As a result, it is difficult to fully gauge how ViMo would perform when integrated with these newer, stronger agents.

**Questions:**

* Could the authors update the experiments by integrating **ViMo** with more recent models such as **UITars** or **Qwen3-VL**, which achieve around 60% accuracy on AndroidWorld?

* With the rapid progress of modern **image and multimodal generation models** (e.g., *Qwen-Image-Edit, Nano-Banana, GPT-Image*), the gap in text rendering and visual fidelity is quickly shrinking. Do the authors envision a future where **digital world models** can be trained **end-to-end**, without the need for modular designs like STR and text-filling stages?

* Regarding the **conceptual validity of digital world models**—humans can intuitively predict outcomes in the *physical* world, but GUI environments are often highly dynamic, with frequent content refreshes and animations that even users cannot anticipate precisely. How do the authors view the **epistemic limits** of modeling such digital environments? Do the authors see ViMo’s predictive modeling as primarily a *functional approximation* for agents, rather than a realistic simulation of human prediction?

---

> ### Author Response · Authors · 2025-11-20
>
> We appreciate the reviewer's time and insightful feedback on our submission. Below, we respond to the concerns raised.
>
> **Comment 1**: Could the authors update the experiments by integrating ViMo with more recent models, such as UITars or Qwen3-VL?
>
> **Response 1** :
>
> Thanks for the suggestions. For UITars, we tested UI-Tars-1.5-7B, which has been trained and rewarded with strong supervision. However effective, such supervision usually fits in a specific interaction pattern, and it does not generalise well to new prompts, new action spaces, or new tasks to tackle real-world situations. Experimentally, it achieves 0\% with T3A prompt and 1.72\% on M3A prompt, on AndroidWorld tasks. A major difference between T3A and M3A is whether the screenshot image is provided to the agent.
>
> Supervision is undoubtedly a stronger form of guidance than world model-predicted future GUI states. However, such supervision is limited in its ability to cover diverse interaction patterns (with or without images, different action spaces, varying prompt structures, and different output formats) and cannot cover all possible scenarios. In contrast, a world model is more suitable for agents whose knowledge is not overly sensitive to a specific domain and instead benefits from having the future GUI as guidance.
>
> For Qwen3-VL, performance of Qwen3-VL-8B-Instruct on the M3A prompt increases from 36.21\% without the world model to 38.79\% after integrating it.
>
> **Comment 2**: With the rapid progress of modern image and multimodal generation models, the gap in text rendering and visual fidelity is quickly shrinking. Do the authors envision a future where digital world models can be trained end-to-end, without the need for modular designs like STR and text-filling stages?
>
> **Response 2**:
>
> Thank you for the question. Although recent multimodal generation models have made impressive progress in visual quality and text rendering, jointly generating GUI graphics and precise text content remains an open challenge.
>
> GUI screenshots are uniquely difficult to synthesise as they contain dense layouts and fine-grained text elements. Rendering GUI text accurately often requires megapixel-level resolution, and even minor pixel-level errors can distort small fonts. This is why ViMo intentionally separates GUI graphic generation from text content generation, and this design choice remains necessary under current model capabilities.
>
> Looking forward, we do envision an end-to-end digital world model, but it would require a generative architecture capable of producing pixel-level GUI graphics and word-level text tokens within a single framework. The key challenge is how to align these two fundamentally different generation processes and train them jointly. Developing such a unified generative model remains an exciting direction for future research.
>
> **Comment 3**:
>  ...GUI environments are often highly dynamic that users cannot anticipate precisely. How do the authors view the epistemic limits of modelling such digital environments?
>
> **Response 3**:
>
> To model GUI environments, one fundamental limitation lies in how much contextual information can realistically be provided and integrated into the generation process. In ViMo, the available context includes the current GUI image, the user goal and the user action, and additional details are not explicitly specified. For example, in Fig. 3, one generated screen shows "5:49" in the top-left corner for the command "set timer for 20 minutes", while another shows "Timer" for "set timer for 12 minutes". Both outcomes are plausible in context, and more importantly, the functional element (the timer) remains consistent with the user action in both cases.
>
> Therefore, we view digital world models as approximate, not exact, representations of the digital environment. Their goal should be to faithfully capture functional, instruction-relevant changes, while allowing reasonable variability in non-essential GUI details.
>
> **Comment 4**:
> Do the authors see ViMo's predictive modelling as primarily a functional approximation for agents, rather than a realistic simulation of human prediction?
>
> **Response4**:
>
> We believe the distinction between "functional approximation" and "realistic simulation" is not strictly separable. In practice, real-world models are motivated to serve both purposes, as applications such as agents are typically optimised on real-world scenarios and perform better when the simulated world reflects a more realistic environment.
>
> Importantly, Table 1 shows that our method achieves a solid improvement on automatic metrics (71.82\%) and an even larger gain under human evaluation (225.93\%). These results indicate that ViMo can approximate a GUI world with a level of realism not demonstrated in prior work, and this realism directly benefits downstream app-agent tasks. ViMo's modelling is therefore both aligned with human prediction and effective as a functional approximation for app agents.

---

> > ### Author Response · Authors · 2025-11-27
> >
> > Dear Reviewer Wy6A,
> >
> > We would like to express our sincere gratitude for your positive and encouraging feedback. We hope that our clarifications and additional explanations are clear and helpful.
> >
> > Best regards

---

### Official Review · Reviewer_yGAq · 2025-11-01

**Soundness:** 3
**Presentation:** 2
**Contribution:** 2
**Rating:** 4
**Confidence:** 3

**Summary:**

This paper proposes ViMo, the first visual GUI world model that predicts future app observations as images rather than text descriptions. The key innovation is the Symbolic Text Representation (STR), which overlays text content with rectangular placeholders to decompose GUI generation into graphics and text generation. ViMo uses a diffusion-based STR Predictor for graphics and an LLM-based GUI-text Predictor for text content. Experiments demonstrate improvements in GUI quality metrics and app agent decision-making performance.

**Strengths:**

**(1) First visual world model formulation for GUI agents:**
ViMo is among the first works to explicitly formulate GUI state prediction as a generative visual world model, extending diffusion-based modeling beyond robotics and embodied domains to mobile GUI environments.

**(2) Novel STR Representation:**
The proposed Symbolic Text Representation (STR) elegantly addresses the challenge of text generation in GUIs by decoupling location prediction (via diffusion) from content generation (via LLM).
This decomposition is technically sound, mitigates the multimodal alignment issue highlighted in Fig. 1, and may generalize beyond GUI generation to other structured visual reasoning tasks.

**(3) Integration with existing App agents and measurable improvements:**
ViMo is integrated as a plug-in to current App agents (T3A, M3A) and achieves consistent step-level accuracy gains (+9–14%) as well as moderate task-level improvement (+7.8%) on AndroidWorld.
These results demonstrate that visual world models can effectively complement decision-making modules within agent pipelines.

**(4) Thorough experimental analysis:**
The paper provides detailed evaluations, including trajectory synthesis, error-accumulation analysis, qualitative failure cases, and inference-time comparison.
This level of transparency and comprehensive reporting is commendable and contributes to the reproducibility of the study.

**Weaknesses:**

**(1) Limited evaluation data scale:**
The main experiments rely on relatively small-scale or sampled settings, including 19 apps with 57 episodes for world-model evaluation and 116 tasks for online testing.
This limited scope raises concerns about statistical significance and makes it difficult to assess robustness across diverse GUI domains and app categories.

**(2) Evaluation method validity concerns:**
Most reported metrics depend on LLM-as-judge evaluations and a small-scale human study (~20 samples per method).
While these methods provide qualitative insights, they introduce potential bias, subjectivity, and circularity, especially since the same class of LLMs is used for both generation and evaluation.

**(3) Modest performance gains with high computational cost:**
As shown in table 12, ViMo achieves 49.20% step accuracy, only 2.56 points higher than the Change-Text baseline, while requiring approximately two minutes per inference, about 24 times slower than the five-second language baseline.
Such computational overhead raises concerns about the model’s practical applicability given the modest improvement in accuracy.

**(4) Limited competitiveness against existing GUI agents in other paradigms:**
Compared with the current AndroidWorld leaderboard (https://docs.google.com/spreadsheets/d/1cchzP9dlTZ3WXQTfYNhh3avxoLipqHN75v1Tb86uhHo/edit?gid=0#gid=0), where most end-to-end or grounding-based agents complete steps in under 10 seconds and achieve up to 97.4% accuracy, ViMo’s 40.95% task accuracy and significantly higher latency make it less competitive.

**(5) Presentation and clarity issues:**
- The paper’s organization is dense and at times difficult to follow.
- Some figures (e.g., Fig. 1 and Fig. 2) are overly compact, and the dense layout and small text reduce overall readability and visual clarity.
- The spacing of several tables (e.g., between Tables 2 and 4–5, and below Table 7) is overly tight, which reduces readability and makes the layout appear visually crowded.
- The appendix is not properly referenced or linked from the main text, making it difficult for readers to locate the corresponding details without manual navigation.

**Questions:**

**(1) Comparison with GUI Agent in other paradigm:**
Given that most end-to-end or grounding-based agents on the current AndroidWorld leaderboard achieve up to 97.4% accuracy with per-step latency below 10 seconds, while ViMo reaches 40.95%, could the authors elaborate on the intended advantages or use cases of adopting a world-model approach in this context?
In particular, under what conditions might such a model complement or outperform grounding-based or end-to-end systems?

**(2) Cost–benefit analysis:**
ViMo achieves only +2.56% improvement over Change-Text while being roughly 24× slower (2 min vs 5 s). For a 10-step task, this translates to about 20 minutes vs 50 seconds.
What use cases justify this trade-off? Have you explored optimizations or hybrid approaches to reduce inference latency?

**(3) Dataset and evaluation scale:**
If only the subset of 57 episodes was used, what criteria were applied when sampling these episodes to ensure representativeness across app types?
Could the authors expand the evaluation to include a larger dataset to strengthen statistical validity and demonstrate robustness across different GUIs?

**(4) Evaluation methodology:**
Since most metrics rely on LLM-as-judge evaluations, have the authors examined inter-rater consistency or any correlation between LLM-based scores and human evaluations?
Such analysis would help validate whether the automatic metrics reliably reflect real-world agent performance.

**(5) Presentation:**
Please consider addressing the issues outlined in the weaknesses section, such as improving figure clarity, table spacing, and better cross-referencing between the main text and appendix.

---

> ### Author Response · Authors · 2025-11-20
>
> We appreciate the reviewer's time and insightful feedback on our submission. Below, we respond to the concerns raised.
>
> **Comment 1**:
> Comparison with GUI Agent in other paradigm: could the authors elaborate on the intended advantages or use cases of adopting a world-model approach in this context (end-to-end agents )? In particular, under what conditions might such a model complement or outperform grounding-based or end-to-end systems?
>
> **Response 1**:
>
> Thanks for the question. End-to-end supervision is undoubtedly a strong form of guidance, with the future implicitly encoded in the supervised trajectories.
>
> However, such systems remain closely tied to the interaction patterns seen during training. Their behaviour can be sensitive to specific prompt formats, action spaces, and input modalities (with or without images), which makes generalisation challenging when any of these conditions shift.
> To illustrate this limitation, we evaluated UI-Tars-1.5-7B under new prompts.
> T3A mainly differs from its original prompt in providing a textual description (no image input), and the result is 0\% on AndroidWorld.
> M3A differs in both the action space and prompt details, and the result is 1.72\%.
> These findings suggest that differences from the original training setup can lead to substantial performance drops.
>
> By contrast, a world model provides task-agnostic GUI transition prediction by modelling how the GUI evolves, and it is independent of prompt style or action representation. This provides a more flexible foundation for agents that operate across varied situations and can benefit from anticipating future states.
> To further validate this, we evaluated Qwen3-VL-8B-Instruct with the M3A prompt on AndroidWorld, and observed an improvement from 36.21\% without the world model to 38.79\% with it.
>
> In summary, world models are most valuable when robustness, cross-task generalisation, or flexibility across prompts and action formats is required. They do not replace end-to-end methods, but they fill an important gap, especially in settings where the model is not overly sensitive to a specific domain.
>
> **Comment 2**:
> ViMo achieves only +2.56\% improvement over Change-Text while being roughly 24× slower (2 min vs 5 s).  What use cases justify this trade-off? Have you explored optimisations or hybrid approaches to reduce inference latency?
>
> **Response 2**:
>
> We appreciate the reviewer’s concerns. Our motivation for using a visual world model is that it provides richer and more detailed future predictions than methods that only update text.  Such a detailed GUI prediction serves as a good intermediate state when the horizon is longer. Table 12 reports results under a 1-step setting and Table 10 evaluates multi-step rollouts (up to 4 steps). Under a horizon of 4 steps, ViMo achieves 12.28\% vs 5.26\% for Change-text, a 2.33 times improvement. This suggests that the benefits of a visual world model become more pronounced as the prediction horizon increases.
>
> Regarding inference efficiency, improved hardware can offer immediate benefits. For example, switching from a V100 to an H200 reduces STR image generation time from about 8 seconds to 4 seconds. Beyond hardware, a promising optimisation direction is to introduce lightweight memory or caching mechanisms: if a prediction condition matches a previously generated one, the system could directly reuse the cached image instead of invoking the full generation pipeline. This would eliminate redundant computation and significantly reduce latency. However, designing an effective and reliable memory management strategy remains an open challenge.
>
> **Comment 3**:
> If only the subset of 57 episodes was used, what criteria were applied when sampling these episodes to ensure representativeness across app types? Could the authors expand the evaluation to include a larger dataset to strengthen statistical validity and demonstrate robustness across different GUIs?
>
> **Response 3**:
>
> We feel there is a misunderstanding. The total episodes were 3550  (L316 and Table 9).
> The dataset was selected for only two reasons in L736-749 in the Appendix:
>
> (1) Noise Filtering.
> We manually filtered out applications in Android in the Wild that contain misaligned trajectories, where the recorded user actions do not correspond to the ground-truth next screens. After this filtering step, 11 clean and reliable apps remained.
>
>  (2) Balancing Training Data.
> The original dataset in Android Control is extremely long-tailed: only 13 out of 759 apps contain more than 50 samples. We therefore focused on these 13 apps to improve learning efficiency and avoid severe imbalance issues.
>
> Regarding evaluation scalability, we also conducted experiments on the full test split of the Android Control dataset (Table 11), which contains 2,855 episodes, approximately 8.2× larger than our test split and 79\% of these episodes were not seen in our dataset, providing a strong measure of generalisation and robustness across diverse GUIs.

---

> ### Author Response · Authors · 2025-11-20
>
> **Comment 4**:
> Evaluation methodology: Since most metrics rely on LLM-as-judge evaluations, have the authors examined inter-rater consistency or any correlation between LLM-based scores and human evaluations? Such analysis would help validate whether the automatic metrics reliably reflect real-world agent performance.
>
> **Response 4**:
>
> For LLM-based consistency, Table 11 reports results from running the generation and evaluation pipeline three independent times, and the standard deviation across runs is very small (0.0025). This indicates that the LLM-as-judge evaluations are internally stable.
>
> Regarding correlations between human preferences and automatic metrics, we observe from Table 1 that
> users tend to give lower scores to models that produce visually unrealistic GUIs, even when the functional content is correct. In contrast, LLM judges focus more on functional correctness and may still assign high scores even if the output does not present an actual GUI. This difference explains the higher scores observed for models such as HTML-vision and UI-diffuser under LLM evaluation.
>
> Overall, LLM-based metrics are stable and reliable for assessing functional correctness, while human evaluations place additional weight on visual realism.
>
> **Comment 5**:
> Please consider addressing the issues outlined in the weaknesses section.
>
> **Response 5**:
>
> Thank you for the suggestions. We will carefully address the issues raised in the weaknesses section.

---

> > ### Author Response · Authors · 2025-11-27
> >
> > Dear Reviewer yGAq,
> >
> > As the discussion period is coming to an end, we would like to ensure that we have fully addressed all of your comments and concerns. If there are any additional points you would like us to clarify or discuss further, we would greatly appreciate the opportunity to do so. Your feedback has been extremely valuable to us, and we remain committed to incorporating your suggestions to further improve our work. We sincerely thank you for the time and effort you have devoted to reviewing our manuscript.
> >
> > We would also like to further clarify our latency costs, as a follow-up to our previous response in **Response 2**. The 2-minute inference time reported in Table 12 includes model loading overhead. Due to limited computational resources, ViMo must be re-initialized for each request, and the model is unloaded after each generation while waiting for the agent to issue the next request, in order to support shared server usage.
> >
> > Under a deployment setting where the model is pre-loaded and kept in memory, the core inference time is approximately 4 seconds for image generation (on an H200 GPU), approximately 30 seconds for text generation (involving 3 sequential GPT calls as discussed in Appendix A.1). With continued advances in large language models, this latency could be further reduced by enabling reliable text generation with less GPT calls.

---

### Official Review · Reviewer_qAJA · 2025-11-02

**Soundness:** 3
**Presentation:** 2
**Contribution:** 3
**Rating:** 6
**Confidence:** 3

**Summary:**

This paper introduces a generative visual world model for mobile app agents that predicts future GUI observations as images rather than textual descriptions. The core idea is to improve long-horizon planning for agents that control mobile apps, by visually simulating the outcomes of user actions. Because rendering text is challenging for diffusion models, the authors decompose generation into graphics (via diffusion) and text (using an LLM). Experiments show that ther proposed method improves the realism of generated GUI screens, particularly the text elements. Using the method for planning improves agent's accuracy (+14%), and generalizes to unseen apps. The overall contribution is a visual diffusion+LLM GUI simulator as an alternative to language-based GUI simulation.

**Strengths:**

- introduces a visual world model that predicts future app GUI observations as images. This is unlike prior language-based world models that generate only text descriptions of GUIs.
- proposes an “STR” representation which overlays all textual content in the GUI with symbolic placeholders (white rectangles), This simplifies the text generation task into text localization while the graphics are generated with a diffusion model
- Proposes two decoupled modules: 1) one generates future GUI structure as an image with placeholders for text (white rectangles) 2) the other one (LLM-based) fills in text content for each placeholder rectangle.
- Good experimental results:
   - Better GUI world model: achieves a significant improvement in automatic GUI quality metrics and in user studies versus existing world models
   - also boosts action prediction accuracy, and, to a lesser degree, online task completion rate

**Weaknesses:**

The paper is generally well written, but there are some sections that are not clear or lack important details
- The dataset section is quite short and does not explain the task that the agent performs
- Why is the dataset subsampled from the larger one? Why not use a standard dataset split?
- Is the 'dataset summarization' referring to training data or test data
- what is "split summarisation"?
- Unclear how successful actions are determined
- Details on how user actions are encoded for the diffusion process are vague

(Minor)
- Table captions do not specify what the columns mean, eg accuracy?
- Typo in "Table 3: Comapre World Models"

**Questions:**

1) Why is the dataset subsampled from the larger one?
This begs the question of whether the test set was selected in such a way as to favor the proposed method over others. Why not use a standard dataset split?

2) Please explain what is the task that the agent must perform
Does the input start with an empty screen? Is the instruction given, e.g. "set an alarm for 9am"?

3) (Most important question) Will this type of approach be practical?

The paper says "Inference time on V100 GPU is 8 seconds on a STR image generation and 30 seconds on GUI-text prediction." However the application is an agent that operates mobile apps. Mobile devices don't have GPUs! How can this method be practically applied? The "online evaluation" implies that the method was applied in real time, but there are very few details.

---

> ### Author Response · Authors · 2025-11-20
>
> We appreciate the reviewer's time and insightful feedback on our submission.
> Below, we respond to the concerns raised.
>
> **Comment 1**: Why is the dataset subsampled from the larger one? This begs the question of whether the test set was selected in such a way as to favor the proposed method over others. Why not use a standard dataset split?
>
> **Response 1**:
>
> To the best of our knowledge, this is the first visual world model that predicts GUI transitions using triplets of (current GUI screenshot, user action, next GUI screenshot). As no existing tasks or datasets directly address this problem, we constructed our dataset by extracting samples from a GUI agent dataset that provides agent trajectories.
> We did not use their entire dataset for two main reasons, as provided in L736-749 in the Appendix:
>
> (1) Noise Filtering.
> We manually filtered out applications in Android in the Wild that contain misaligned trajectories, where the recorded user actions do not correspond to the ground-truth next screens. After this filtering step, 11 clean and reliable apps remained.
>
>  (2) Balancing Training Data.
> The original dataset in Android Control is extremely long-tailed: only 13 out of 759 apps contain more than 50 samples. We therefore focused on these 13 apps to improve learning efficiency and avoid severe imbalance issues.
>
> Importantly, the test set was selected using the same filtering and balancing criteria and was not chosen to favor our method in any way. Since our dataset is derived from an existing and independently collected GUI agent dataset, our processing involved only task-agnostic filtering. The training, validation, and test sets were randomly split from the dataset and were not manually selected or curated to benefit our model.
>
> Furthermore, we would like to emphasise that this curated dataset strengthens the contribution of our paper, as it represents the first high-quality world-model dataset for GUI transition prediction and required substantial human effort to ensure its reliability.
>
> **Comment 2**: Please explain what is the task that the agent must perform. Does the input start with an empty screen? Is the instruction given, e.g. "set an alarm for 9am"?.
>
> **Response 2**:
>
> The agent is given the environment (current GUI screenshot) and a user goal, and must predict the action to execute on the current GUI (L115-118).  The agent is not constrained to start from a specific screen. In the dataset, trajectories often begin from the home screen, but some trajectories also start from within other applications. The agent's objective is to complete the user goal regardless of the initial screen.
>
> In addition, we have provided the dataset information (all train, validation and test splits) in the submitted code to help reviewers better understand the form of the user goal.
>
> **Comment 3**: Will this type of approach be practical. Mobile devices don't have GPUs, How can this method be practically applied.
>
> **Response 3**:
>
> As discussed in Line 420, ViMo is designed to run as a remote API service deployed on a GPU server. Mobile devices interact with the model by sending requests to this server, rather than running the model locally. This architecture is practical because the computationally expensive components run on the server, while the mobile device only performs lightweight communication. This deployment paradigm is widely used in industrial applications, for example, systems like ChatGPT and other large AI services do not execute their models on the user's device, but instead rely on cloud-based inference.
>
>
> **Comment 4**:
> Details for online evaluation..unclear how successful actions are determined
>
> **Response 4**:
>
> Thank you for the question. In our setting, online evaluation means that the agent receives real-time feedback from the environment during execution. The environment we use is the AndroidWorld simulator. In online evaluation, actions are not evaluated individually; instead, the evaluation is based on whether the environment reaches the task-completed state (L432). AndroidWorld also provides a hand-crafted policy that checks the environment state and automatically determines whether the task has been successfully completed.
>
> This differs from offline evaluation, where trajectories are static and success is defined by matching the predicted actions to recorded ground-truth actions. Following Android Control, we determine whether a predicted action matches the ground truth using the following criteria:
> (1) Action type matching: We compare the action type by matching the action-type strings (e.g., click, type_text, scroll).
> (2) Parameter matching: For click actions, we check whether the predicted click location lies inside the bounding box of the target UI element. For actions with parameters (such as text input), we compare the predicted parameter value with the ground-truth parameter.

---

> ### Author Response · Authors · 2025-11-20
>
> **Comment 5**:
> Is the 'dataset summarization' referring to training data or test data.
> What is "split summarisation"?
>
> **Response 5**:
>
> The "dataset summarisation" refers to the whole dataset before any splitting. The "split summarisation" refers to the statistics of the train, validation, and test splits, which are provided in Table 8 (Lines 702–709 in the Appendix).
> To be specific, the dataset is randomly divided into train, validation, and test splits with an 8:1:1 ratio.
>
>
> **Comment 6**:
> Details on how user actions are encoded for the diffusion process are vague
>
> **Response 6**:
>
> Our method uses a U-Net architecture (L232–241) in which user actions and GUI images are jointly integrated into the diffusion process. Specifically, the user action is taken as a text prompt and encoded by the text encoder. During denoising, the U-Net incorporates both the encoded user action and the GUI image features through cross-attention, enabling the diffusion model to condition its prediction on both the user action and the current GUI image.
>
> **Comment 7**:
> Minor questions on Table captions and Typo.
>
> **Response 7**:
>
> Thank you. We will add the missing clarifications to the table captions and fix the typo.

---

> > ### Comment · Reviewer_qAJA · 2025-11-22
> >
> > I have read the authors' rebuttal. Thanks for answering my questions, it clarifies things. I am ok with the paper being accepted provided the clarifications regarding the dataset etc. are made in the final version.
> >
> > I am not convinced that this method is practical. Even if the model runs on a server and the mobile device just sends requests, "Inference time on V100 GPU is 8 seconds on a STR image generation and 30 seconds on GUI-text prediction." Waiting that long might be ok if you're asking ChatGPT to book a flight for you, but we're talking about doing things on your device like setting an alarm. Maybe the authors could add a discussion about which mobile phone tasks could be amenable to this solution.

---

> > > ### Author Response · Authors · 2025-11-23
> > >
> > > We sincerely thank the reviewer for reading our rebuttal and for the positive assessment of our work. We also appreciate the important and thoughtful concerns regarding the practicality of our method.
> > >
> > > Building on our previous discussion of mobile deployment, we would like to provide further analysis and clarification regarding latency. We agree that users might have different latency expectations depending on task complexity: complex tasks, such as making a booking, can tolerate more processing time, while simpler tasks, such as setting an alarm, require faster responses. We acknowledge that current GUI agents still struggle to achieve both low latency and high accuracy for industrial-level deployment. Our core design philosophy prioritises foundational reliability and higher accuracy, especially in scenarios requiring long-horizon planning.  For example, when the planning horizon extends to 4 steps, our visual generation pipeline achieves 2.33× higher accuracy (12.28% vs 5.26% in Table 10) compared to Change-text. We believe that providing a visual world model capable of accurately predicting action outcomes is a crucial and necessary step forward for the agent community.
> > >
> > > Regarding latency optimisation, we note that our STR image generation time can be significantly reduced from 8 seconds to 4 seconds by upgrading computational resources from V100 to H200 GPUs. We believe that hardware improvements and future model optimisation will further narrow this gap, making high-accuracy GUI agents increasingly practical for everyday use.
> > >
> > > We hope this response addresses the reviewer’s concerns regarding practicality and clarifies our future direction toward achieving both high accuracy and deployment feasibility.

---

### Comment · Area_Chair_hHwP · 2025-11-24

Dear authors,

After reviewing the paper and the reviewer responses, I would like to point to these weaknesses to give you an opportunity to respond (some of which partly overlap with reviewer responses, but which I would like a direct response to):

The arguments of this paper says that it provides two key benefits to understanding and operating on mobile phone applications. First, they provide a privacy-preserved training environment that given an GUI and an action they would be able to generate the screen produced by that action, thereby enabling the training of a model to interact with that app to complete tasks. Second, that this world model provides a better representation with with to perform actions, as the representations would have some additional information about the future GUI states.

One argument for rejection observes that the app environment produced by the paper has little utility, as mobile app data is simple to collect. Large scale datasets have been produced for years [A,B], and, thus, this point has merit.  The authors can argue that their world environment has other benefits, such as ensuring a user's data is kept private. However, that has two issues. First, a generative model must be trained on some data, which means that it could contain private information and could leak it in any future generations. Second, these emulators can be setup with generic credentials (as done in [A]), i.e., no private information would be collected.

This paper would still have merit beyond this as it provides beneficial representations on downstream applications. However, in this regard, the paper is missing key comparisons that use language rather than GUIs to help better understand current or future states [B,C]. In fact, [C] is motivated in much the same way as this submission, where its goal is to provide insight into future states. However, it accomplishes this by generating a caption that summarizes the future state rather than the GUI. In this, the papers are clearly different, but the issue is that the authors did not compare to them. As generating a caption is much more lightweight than an image, it would have efficiency benefits over the proposed approach. Regardless, comparing to these closely related works [B,C] is clearly needed to validate the benefits on downstream applications of this paper.

This suggests that both the world model and the application of it have significant concerns.

[A] Rico: A Mobile App Dataset for Building Data-Driven Design Applications. UIST 2017

[B] Spotlight: Mobile UI Understanding using Vision-Language Models with a Focus. ICLR 2023

[C] Tell Me What's Next: Textual Foresight for Generic UI Representations. Findings of the ACL, 2024

---

> ### Author Response · Authors · 2025-11-25
>
> We thank the Area Chair hHwP for the opportunity to respond to these important concerns. Below, we address each point in detail.
>
> (a) Mobile data is easy to collect.
>
> Static mobile data is not ready to reflect **dynamic** user actions and mobile environment. For example, user actions, such as the one illustrated in Fig. 1 ("Enter the email as dbwscratch.test.id5@gmail.com"), involve unpredictable and instance-specific information that cannot be reliably represented by a purely static dataset. Therefore, our goal is to generate the next GUI state given a current GUI screenshot and a user action. This task requires capturing dynamic user interactions in order to accurately model how the interface changes after an action is applied.
>
> (b) Potential leakage of user data during training.
>
> We address the concern of potential user data leakage by analysing the two modules of our method separately.
>
> For GUI graphic generation, the diffusion model is **trained** on our **STR representation (GUI graphic)**, in which all textual content is **removed** from the GUI. The model is trained solely to predict visual and layout information. As a result, no sensitive or personally identifiable information is included in the training data or generated by the model.
>
> For text rendering, no model fine-tuning is performed. At **test time only**, an LLM (such as ChatGPT) is prompted to generate textual content based on contextual information such as the previous GUI state, the user action, and the user goal. Sensitive user information that is not displayed on the screen is **never seen or accessed**. Sensitive information that may appear on screen (e.g., contact lists or account details) is **not collected, stored, or reused for training** or any other purpose.
>
> Therefore, no private user information is collected, stored, or leaked during either the training or inference stages.
>
> (c) Comparison with language models.
>
> We compare our method with language-based baselines under two settings (reported in Tables 3, 9, and 10):
>
> (1) Change-text, where an LLM is prompted to summarise how the GUI will change after a user action is applied to the current screenshot.
>
> (2) HTML-text, where an LLM is used to predict the next app observation in the HTML format.
>
> In the Change-text setting, the interaction protocol is consistent with Textual Foresight [C]: the model takes as input the current GUI image and the user action and outputs a textual summary.
> To showcase their similar task objectives, we provide a qualitative example comparing outputs from the Change-Text and Textual Foresight [C], based on the example shown in Fig. 1 of [C].
>
> Change-Text (our baseline):
> A context menu (bottom sheet or popup) opens with options for the selected song, such as Add to playlist, Like/Save, Share, Go to artist, and View album.
>
> Textual Foresight [C]:
> Song information and options for playing, saving, sharing, reporting explicit content, and viewing credits.
>
> **Summary:**
>
> (1) ViMo focuses on dynamic GUI transitions, rather than static mobile data.
>
> (2) Sensitive user data is not used in training.
>
> (3) We conducted fair comparisons with language-based models, and our Change-text baseline shares the same task objective with [C].
>
> We again thank the Area Chair hHwP for raising these valuable points. We will clarify and expand the discussion of  the suggested related works [a,b,c] in the revised manuscript.

---

> ### Comment · Area_Chair_hHwP · 2025-11-26
>
> I have two follow up questions:
>
> 1. Was the LLM you used for the text change baseline trained for the task at all? If not, you should comment on the fairness of your experiments.  Specifically, it sounds like you trained an image generator to produce your target GUIs, so not doing so for the LLM could explain any performance differences
>
> 2. What is the inference time differences between the two methods?
>
> I shall also remind the authors they can modify their paper during this rebuttal process.  Providing a revised paper can help showcase your intended changes and alleviate concerns about them.

---

> > ### Author Response · Authors · 2025-11-27
> >
> > Thanks Area Chair  hHwP for the follow up questions. Below, we address each point in detail.
> >
> > 1) Was the LLM used for the text change baseline trained for the task at all?
> >
> > The LLM used in our Change-text baseline is GPT-4o, a strong general-purpose large language model with broad knowledge.  In our experiments, GPT-4o was used in an off-the-shelf manner, without any task-specific fine-tuning. Importantly, the same model (GPT-4o) was also used in our GUI text generation module, ensuring consistency and a fair comparison in Tables 3,9 and 10.
> >
> > 2) Comment on the fairness of the experiments.
> >
> > To evaluate whether the vision modality provides more precise information than the language modality for app agent tasks, we conduct controlled comparisons under a non-fine-tuned setting. As shown in Tables 3, 9, and 10, our two visual baselines (HTML-vision and UI-diffuser), which are not fine-tuned on our task, consistently outperform the language-based methods (Change-text and HTML-text), which are also not fine-tuned. This result reflects the inherent advantage of visual representations for modelling fine-grained GUI details.
> >
> > To evaluate the necessity of our ViMo design, we also fine-tune the vision-based baseline IP2P using the same training data as ViMo. This ensures a fair comparison and further verifies that the observed improvements stem from our model architecture rather than advantages in the training data.
> >
> > 3) What is the inference time differences between the two methods?
> >
> > As reported in Table 12, under our experimental setup, the total inference time per request for ViMo is approximately 2 minutes, including model loading and communication overhead, while Change-text requires approximately 5 seconds. A more detailed breakdown shows that our STR image generation stage takes around 8 seconds on a V100 GPU, and the text generation stage takes approximately 30 seconds.
> > Although our method introduces additional latency, this is an intentional design choice that is closely aligned with our objective of improving the reliability of long-horizon decision making. For example, when the planning horizon is extended to 4 steps, our ViMo pipeline achieves 2.33× higher accuracy (12.28% vs. 5.26% in Table 10) compared to Change-text.
> > We view the ability to construct a visual world model that can accurately predict the consequences of actions as a critical foundation for building robust GUI agents. While this design involves additional computational cost, we believe this trade-off is justified by the substantial gains in reliability and planning accuracy, especially when the horizon is longer.
> >
> > Furthermore, the STR image generation latency can be reduced from 8 seconds to 4 seconds by upgrading the hardware from V100 to H200 GPUs. We expect that continued advances in hardware and model optimization will further narrow this gap, making high-accuracy GUI agents increasingly practical. This also represents an important future direction toward achieving both high accuracy and real-world deployability.
> >
> > We will revise the manuscript to clarify discussed points in the next version before 29th Nov.

---

> > > ### Comment · Area_Chair_hHwP · 2025-11-27
> > >
> > > Did you explore methods of making the inference time more fair? It sounds like even large ensembles would be much faster as is commonly known to boost performance.

---

> > > > ### Author Response · Authors · 2025-11-27
> > > >
> > > > We respond to this question as follows.
> > > >
> > > > (1) Did you explore methods of making the inference time more fair?
> > > >
> > > > Our current inference pipeline consists of two main components: (a) Approximately 4 seconds for  STR GUI image generation, on H200 GPU.  (b) Approximately 30 seconds for text generation, which involves three sequential GPT calls and local image post-processing.
> > > >
> > > > We initially experimented with a single-pass GPT call to generate all textual content, which required only about 5 seconds. However, we found that a single call was not sufficiently reliable to handle this complex task of predicting textual content in the generated STR image. As a result, we decomposed the process into three sub-tasks (described in Appendix A.1), which currently necessitates three sequential calls to achieve stable and reliable performance.
> > > > We would also like to clarify that the 2-minute inference time reported in Table 12 includes model loading overhead. Due to our limited computational resources, ViMo must be re-initialized for each request, and the model is unloaded after each generation  while waiting for the agent to issue the next request, to support shared server usage.  Under a deployment setting where the model is pre-loaded and kept in memory, the core inference time is approximately 4 seconds for image generation and 30 seconds for text generation. With continued improvements in large language models, this latency could be further reduced by enabling reliable text generation with less GPT calls.
> > > >
> > > > (2) It sounds like even large ensembles would be much faster as is commonly known to boost performance.
> > > >
> > > > We agree that large ensembles of agents can improve performance, particularly through strategies such as majority voting. However, ensembling existing agents does not address the gap of long-horizon GUI state transitions, which is the focus of our work. These represent complementary but distinct problem settings, and both directions are valuable.

---

> > > > > ### Author Response · Authors · 2025-11-28
> > > > >
> > > > > Dear Area Chair hHwP,
> > > > >
> > > > > Thank you for your suggestions to improve our paper. We have prepared a revised version in which all modifications are highlighted in blue.
> > > > >
> > > > > The main changes are summarised below:
> > > > >
> > > > > a) Added a Discussion section in the Appendix D to address:
> > > > >
> > > > > * user privacy preservation
> > > > >
> > > > > * fairness comparisons
> > > > >
> > > > > b) Expanded the Limitations and Future Work section in Appendix E.
> > > > >
> > > > > c) Added a discussion of practical considerations in Appendix C and execution inference time for different modules in Table 12.
> > > > >
> > > > > Additional updates based on reviewers’ comments:
> > > > >
> > > > > a) All appendices are now referenced more clearly in the main text.
> > > > >
> > > > > b) Figures 1 and 2 are updated for better clarity.
> > > > >
> > > > > c) Table layouts are improved to enhance readability.
> > > > >
> > > > > The revised paper now exceeds 9 pages but remains within the 10-page limit for rebuttal submission.

---

### Meta-Review · Area_Chair_AKz9 · 2026-01-03

**Summary:**

This paper introduces a generative visual world model for mobile app agents that predicts future GUI observations as images rather than textual descriptions. It gets 6, 4, 6, 8 in the first round. The main concerns are writting problems, limited evaluation, high computational cost, experimental setup. In rebuttal, most of these concerns are addressed.  I am leaning to accept this paper. Author should revise the paper according to discussion.

**Reviewer Concerns:**

Concerns of all reviewers are addressed in the rebuttal.

**Reviewer Scores:**

Reviewer qAJA would not change their score.
Reviewer yGAq may change their score to 6.
Reviewer Wy6A would not change their score.
Reviewer udhG would not change their score.

---

### Decision · Program_Chairs · 2026-01-26

Accept (Poster)